# An Updated Parameterization of the Unstable Atmospheric Surface Layer in the WRF Modeling System

Prabhakar Namdev[1], Maithili Sharan[1], Piyush Srivastava[2], Saroj K. Mishra[1]

[1]Centre for Atmospheric Sciences, Indian Institute of Technology Delhi, New Delhi, 110016, India

[2]Centre of Excellence in Disaster Mitigation and Management, Indian Institute of Technology Roorkee, Roorkee, 247667, India

*Correspondence to*: Prabhakar Namdev (Prabhakarnmdv587@gmail.com)

**Abstract.** Accurate parameterization of atmospheric surface layer processes is crucial for weather forecasts using numerical weather prediction models. Here, an attempt has been made to improve the surface layer parameterization in the Weather Research and Forecasting Model (WRFv4.2.2) by implementing similarity functions proposed by Kader and Yaglom (1990) to make it consistent in producing the transfer coefficient for momentum observed over tropical region (Srivastava and Sharan 2015). The surface layer module in WRFv4.2.2 is modified in such a way that it contains the commonly used similarity functions for momentum ($\varphi_m$) and heat ($\varphi_h$) under convective conditions instead of the existing single functional form. The updated module has various alternatives of $\varphi_m$ and $\varphi_h$, which can be controlled by a flag introduced in the input file. The impacts of utilizing different functional forms have been evaluated using the bulk flux algorithm as well as real-case simulations with the WRFv4.2.2 model. The model-simulated variables have been evaluated with observational data from a flux tower at Ranchi (23.412N, 85.440E; India) and the ERA5-Land reanalysis dataset. The transfer coefficient for momentum simulated using the implemented scheme is found to agree well with its observed non-monotonic behaviour in convective conditions (Srivastava and Sharan, 2021). The study suggests that the updated surface layer scheme performs well in simulating the surface transfer coefficients and could be potentially utilized for parameterization of surface fluxes under convective conditions in the WRF model.

## 1 Introduction

Inadequate representation of near-surface turbulent processes adds significant uncertainty in both climate projections and seasonal weather forecasts obtained from atmospheric models (Bourassa et al., 2013). Most of the numerical weather prediction and general circulation models utilize Monin-Obukhov similarity theory (MOST; Monin and Obukhov 1954) to parameterize surface turbulent fluxes. To estimate these fluxes and near-surface atmospheric variables, the theory utilizes similarity functions of momentum ($\varphi_m$) and heat ($\varphi_h$) often prescribed as functions of $\zeta$ (stability parameter). However, the exact functional forms for these functions have not been provided by MOST, rather it suggests some asymptotic predictions under near-neutral to very stable and unstable conditions. Over the years, researchers have developed many functional forms for these functions based on the different experiments, conducted over different locations and have separate expressions for stable

and unstable stratifications (Webb, 1970; Businger, 1971; Carl et al., 1973; Dyer, 1974; Hicks, 1976; Holtslag and De Bruin,
1988; Brutsaert, 1992; Bruin, 1999; Wilson, 2001; Cheng & Brutsaert, 2005; Grachev et al., 2007; Gryanik et al. 2020;
Srivastava et al. 2020).

In most of the atmospheric models, the commonly used similarity functions under convective conditions are those

proposed by Businger (1966) and A. J. Dyer [1965, unpublished work; see Businger (1988)] and referred to as Businger-Dyer
(BD) functions. However, these functional forms are unable to follow the classical free convection limit. The study by Rao et
al. (1996) suggests that the MOST using Businger relations is unable to define transfer coefficient for momentum ($C_D$)
consistent with its observed behaviour, specifically at low wind convective conditions, indicating that MOST needs to be
modified in the (nearly) windless free convection limits. As a result, a revised scaling of heat flux for weakly forced convection
in the atmosphere has been proposed by Rao et al. (2006). Later, the issues of using BD functions in the surface layer scheme
based on the fifth-generation Pennsylvania State University-National Centre for Atmospheric Research Mesoscale Model
(MM5) of a regional scale model (Weather Research and Forecasting; WRF) have been reported in a study by Jimenez et al.
(2012). They implemented the new scheme (referred to as revised MM5 scheme; Jimenez et al., 2012) in the WRF modeling
system and replaced the BD functions by those proposed by Fairall et al. (1996) (F96) under convective conditions. F96
functions are the combination of BD functions, and the functions suggested by Carl et al. (1973) and are valid for the entire
range of atmospheric instability. Note that the most recent version of the WRF model still utilizes F96 functions under
convective conditions.

Srivastava and Sharan (2015) analyzed the observed behaviour of $C_D$ over an Indian land surface and suggested that

the observed $C_D$ shows non-monotonic behaviour with $-\zeta$, unlike the behaviour of predicted $C_D$ from MOST based
parameterization using commonly used $\varphi_m$ and $\varphi_h$ (Businger et al., 1971; Carl et al., 1973; Fairall et al., 1996). Later, a
theoretical study by Srivastava and Sharan (2021) revealed that the three-sublayer model based on Kader and Yaglom (1990)
is able to predict $C_D$ consistent with its observed non-monotonic behaviour. Note that the three-sublayer model has not yet
been newly installed and evaluated in the WRF modeling framework. However, it is already being operational in the surface
layer scheme (Community Land Model; CLM) of National Centre for Atmospheric Research Community Atmosphere Model
version 5 (NCAR-CAM5) as well as Regional Climate Model (RegCM).

The study by Srivastava and Sharan (2021) also analyzed the possible uncertainties associated with the use of different

functional forms of $\varphi_m$ and $\varphi_h$ under convective conditions. To quantify the impacts of different functional forms, they
classified available $\varphi_m$ and $\varphi_h$ in four classes based on the exponents appearing in the expressions of $\varphi_m$ and $\varphi_h$ as (1)
functional forms having the exponents of $\varphi_m$ and $\varphi_h$ as $-1/4$ and $-1/2$, respectively (Businger et al. 1971; Hogstrom 1996).
(2) functional forms having the exponent of $\varphi_m$ and $\varphi_h$ as $-1/3$ (Carl et al. 1973). (3) functional forms having the exponent
of $\varphi_m$ and $\varphi_h$ as $-1/4$ and $-1/2$, respectively in near-neutral conditions while $-1/3$ in very unstable conditions (Fairall et
al. 1996; Grachev et al. 2000; Fairall et al. 2003). (4) functional forms having the exponent of $\varphi_m$ and $\varphi_h$ as $-1/4$ and $-1/2$,
respectively in near-neutral conditions however, $1/3$ for $\varphi_m$ and $-1/3$ for $\varphi_h$ in strong unstable conditions (Kader and
Yaglom 1990; Zeng et al. 1998). This study concluded that utilizing different functional forms of similarity functions in the
bulk flux algorithm results in a large deviation in the values of estimated fluxes. The detailed description of different functional
forms for $\varphi_m$ and $\varphi_h$ considered in different classes are given in Appendix A. We wish to highlight that all available functional
forms for $\varphi_m$ and $\varphi_h$ under convective conditions fall in one of the classes stated above.
The revised MM5 surface layer scheme of the WRF model version 4.2.2 (WRFv4.2.2) employed $\varphi_m$ and $\varphi_h$ based
on Fairall et al. (1996), which belong to class 3. As a result, this scheme is not appropriate in producing $C_D$ consistent with its
observed behaviour, specifically over the Indian land as stated above. Recently Namdev et al. (2023) argue that the
performance of NWP models varies a lot over different seasons and surface types depending upon the functional behaviour of
$\varphi_m$ and $\varphi_h$. Thus, to enhance the potential applicability of the WRF modeling framework, this study attempted to incorporate
all the commonly used similarity functions under convective conditions along with KY90 as well as existing functional forms
in the revised MM5 surface layer scheme of WRFv4.2.2. A namelist flag has been introduced in WRF model to choose between
various $\varphi_m$ and $\varphi_h$ in the modified scheme. The modified surface layer scheme proposed in this study has been evaluated
using offline simulations with bulk flux algorithm as well as the real-case simulations with WRFv4.2.2 during the pre-monsoon
season (March-April-May) of 2009 over a domain centered around the location of the flux tower installed at Ranchi (23.412N,
85.440E), India.
**2 Methodology and data**
**2.1 Surface flux computation in the WRF modeling system**
The Monin-Obukhov similarity theory serves as the foundation for the surface layer parameterization (revised MM5 scheme)
in the WRF model, and the surface turbulent fluxes are calculated based on the bulk approach using bulk transfer coefficients
for momentum ($C_D$) and heat ($C_H$) (Namdev et al., 2024; Srivastava et al., 2021; Srivastava and Sharan, 2021). Following
MOST they are formulated as follows:
$$C_D = k^2 \left[ \ln\left(\frac{z + z_0}{z_0}\right) - \left\{ \psi_m\left(\frac{z + z_0}{L}\right) - \psi_m\left(\frac{z_0}{L}\right) \right\} \right]^{-2} \qquad (1)$$
$$C_H = k^2 \left[ \ln\left(\frac{z + z_0}{z_0}\right) - \left\{ \psi_m\left(\frac{z + z_0}{L}\right) - \psi_m\left(\frac{z_0}{L}\right) \right\} \right]^{-1} \left[ \ln\left(\frac{z + z_h}{z_h}\right) - \left\{ \psi_h\left(\frac{z + z_h}{L}\right) - \psi_h\left(\frac{z_h}{L}\right) \right\} \right]^{-1} \qquad (2)$$
in which k is the von Karmann constant; $z_0$ and $z_h$ are the roughness lengths for momentum and heat, respectively; $\psi_m$ and
$\psi_h$ are the integrated similarity functions for momentum and heat, respectively; and L is the Obukhov length scale.
Their determination based on MOST using integrated forms of the similarity functions is explained in Appendix B. In the
following, the default similarity functions used in WRF are explained and other functions are introduced in Section 2.2.
The default version of the revised MM5 scheme in the WRF model utilizes similarity functions suggested by Cheng
and Brutsaert (2005) under stable atmospheric conditions ($\zeta > 0$), which are developed using the CASES-99 dataset. The
integrated forms of functions proposed by Cheng and Brutsaert are
$\quad \psi_m(\zeta) = -a \ln\left(\zeta + [1 + \zeta^b]^{1/b}\right), \qquad \zeta > 0$ (3)
$\quad \psi_h(\zeta) = -c \ln\left(\zeta + [1 + \zeta^d]^{1/d}\right), \qquad \zeta > 0$ (4)
$\quad$ where $d = 1.1, c = 5.3, b = 2.5$ and $d = 6.1$.
$\qquad$ On the other hand, the similarity functions for unstable atmospheric surface layer ($\zeta < 0$) are those proposed by
$\quad$ Fairall et al. (1996; F96). The corresponding integrated functional forms $\psi_m$ and $\psi_h$ are defined as:
$\quad \psi_\alpha(\zeta) = \dfrac{\psi_{\alpha_{BD}}(\zeta) + \zeta^2\,\psi_{\alpha_{conv}}(\zeta)}{1 + \zeta^2}, \qquad \alpha = m, h.$ (5)
$\quad$ where $\psi_{\alpha_{BD}}$ and $\psi_{\alpha_{conv}}$ denote the integrated functional forms based on Businger and Dyer, and Carl et al. (1973),
$\quad$ respectively. The expressions for $\psi_{\alpha_{BD}}$ and $\psi_{\alpha_{conv}}$ are
$\quad \psi_{m_{BD}}(\zeta) = 2\ln\left(\dfrac{1+x}{2}\right) + \ln\left(\dfrac{1+x^2}{2}\right) - 2\tan^{-1}x + \dfrac{\pi}{2},$ (6)
$\quad \psi_{h_{BD}}(\zeta) = 2\ln\left(\dfrac{1+x^2}{2}\right),$ (7)
$\quad$ in which $x = (1 - 16\zeta)^{1/4}$ and
$\quad \psi_{\alpha_{conv}} = \dfrac{3}{2}\ln(y^2 + y + 1/3) - \sqrt{3}\tan^{-1}(2y + 1/\sqrt{3}) + \dfrac{\pi}{\sqrt{3}}$ (8)
$\quad$ with $y = [1 - \beta_{m,h}\zeta]^{1/3}$. The values of the constants $\beta_m$ and $\beta_h$ are taken as 10 and 34 based on Grachev et al. (2000).
**2.2 Implementation of different similarity functions**
$\quad$ In this section, we briefly describe the implementation of different similarity functions for unstable stratification in the surface
$\quad$ layer parameterization of WRFv4.2.2. Note that two sets of functional forms, namely those suggested by Carl et al. (1973) and
$\quad$ the three sub-layer model proposed by Kader and Yaglom (1990) for convective conditions have not been included and tested
$\quad$ in the surface layer scheme of the WRF modeling framework.
**2.2.1 Functions by Businger et al. (1971) (BD71)**
$\quad$ Similarity functions suggested by Businger et al. (1971) are based on the KANSAS dataset (Izumi, 1971). These functions do
$\quad$ not satisfy the classical free convection limit as predicted by the MOST. They are already implemented in the old version of
the MM5 surface layer scheme (Grell et al., 1994) in the WRF model. The integrated functional forms ($\psi_m$ and $\psi_h$) for $\varphi_m$
and $\varphi_h$ stated in Eqns. (A1) and (A2) (Appendix A) are given in Eqns. (6) and (7).

## 2.2.2 Functions by Carl et al. (1973) (CL73)

Carl et al. (1973) proposed an expression of similarity functions $\varphi_m$ and $\varphi_h$ valid for the stability range $-10 \leq \zeta \leq 0$. The
expressions for $\varphi_m$ and $\varphi_h$ are given in Eqns. (A3) and (A4) (Appendix A). The similarity functions proposed by Carl et al.
(1973) have not been analyzed in the surface layer scheme of the WRF model. The integrated forms ($\psi_m$ and $\psi_h$) of similarity
functions $\varphi_m$ and $\varphi_h$ are given by Eqn. (8).

## 2.2.3 Functions by Kader and Yaglom (1990) (KY90)

Kader and Yaglom (1990) introduced a three-sublayer model for convective conditions. The three sublayers are categorized
based on $\zeta$ values as (1) the dynamic sublayer which corresponds to near-neutral conditions, (2) the dynamic convective
sublayer which corresponds to moderately unstable conditions and (3) the free convective conditions. The present study
utilized $\varphi_m$ and $\varphi_h$ expressions given in Eqns. (A9), and (A10) (Appendix A) that are being used in the surface layer scheme
(CLM4.0; Zeng et al. 1998) of NCAR-CAM5 model. The corresponding integrated forms for $\varphi_m$ and $\varphi_h$ are
$$\psi_m(\zeta) = \begin{cases} \psi_{m1}(\zeta_m) + \ln\frac{\zeta}{\zeta_m} - 1.14\left[(-\zeta)^{1/3} - (-\zeta_m)^{1/3}\right], & \zeta \leq -1.574(= \zeta_m) \\ \psi_{m1}(\zeta) = 2\ln\left(\frac{1+x}{2}\right) + \ln\left(\frac{1+x^2}{2}\right) - 2\tan^{-1}x + \frac{\pi}{2}, & -1.574 < \zeta < 0 \end{cases} \qquad (9)$$

$$\psi_h(\zeta) = \begin{cases} \psi_{h1}(\zeta_h) + \ln\frac{\zeta}{\zeta_h} - 0.8\left[(-\zeta)^{-1/3} - (-\zeta_h)^{-1/3}\right], & \zeta \leq -0.465(= \zeta_h) \\ \psi_{h1}(\zeta) = 2\ln\left(\frac{1+x^2}{2}\right), & -0.465 < \zeta < 0 \end{cases} \qquad (10)$$

where $x = (1 - 16\zeta)^{1/4}$.
Note that all the functions stated above have been newly installed in the revised MM5 surface layer scheme of
WRFv4.2.2 and can be used in place of F96 functions already employed in the model. Here, we have introduced a new surface
layer module where different options for $\varphi_m$ and $\varphi_h$ can be controlled using an appropriate value of namelist parameter
(psimhu_opt). The parameter psimhu_opt is added under the physics section of the namelist file. The variable psimhu_opt can
have values 0, 1, 2, and 3 for different options for functions F96 (default), BD71, CL73, and KY90, respectively. A brief
structure and different choices for psimhu_opt based on newly installed and default functional forms of $\varphi_m$ and $\varphi_h$ in the
default and modified revised MM5 scheme are shown in Figure 1.

## 2.3 Characteristics of default and newly installed similarity functions

The expressions of $\varphi_m$ and $\varphi_h$ for different functional forms utilized in this study are stated in Appendix A. Figure S1 (supplementary material) shows the variation of different (a) $\varphi_m$ and (b) $\varphi_h$ under moderately to strongly unstable conditions. It is evident from Figure S1 that all the different functional forms provide similar values of $\varphi_m$ and $\varphi_h$ in near-neutral to moderately unstable conditions (up to $\zeta = -0.1$ approximately). However, at higher instabilities one can expect noticeable differences between different functional forms of $\varphi_m$ and $\varphi_h$. Note that the functional forms for $\varphi_m$ corresponding to BD71 and CL73 decrease continuously on increasing instability; however, $\varphi_m$ corresponding to KY90 functional forms show decreasing behaviour in near-neutral to moderately unstable conditions and attain a minimum at $\zeta = -1.574$, and, as the instability further increases, it starts increasing with $-\zeta$ (Figure S1a). This implies that $\varphi_m$ based on class 4 functions shows non-monotonic behaviour which contradicts the classical MOST prediction. On the other hand, in case of $\varphi_h$, all the functional forms provide continuously decreasing behaviour of $\varphi_h$ from near-neutral to moderately unstable conditions (Figure S1b).

Figure 2 illustrates the variation of default (F96) and newly installed integrated similarity functions $\psi_m$ and $\psi_h$ (BD71, CL73, and KY90) with respect to $-\zeta$. One can see from Figure 2a that $\psi_m$ corresponding to F96, BD71, and CL73 functional forms increases continuously with $-\zeta$ in moderately to strongly unstable conditions. However, a non-monotonic behaviour has been found for $\psi_m$ corresponding to the KY90 functions implying it first increases with $-\zeta$ and reaches a maximum at $\zeta = -1.574$ and then starts decreasing as instability further grows. On the other hand, $\psi_h$ corresponding to all the considered functional forms increases continuously in near-neutral to strong unstable conditions. However, the rate of increase is slightly higher for F96 in comparison to the other three functions (BD71, CL73, and KY90), whose results are very similar to each other (Fig. 2b).

## 2.4 Observational data for model evaluation

For the evaluation of different simulations corresponding to newly installed similarity functions, observational data derived from the micrometeorological tower installed at Ranchi (India) has been utilized (Srivastava and Sharan, 2019; Srivastava et al., 2020; 2021). The dataset (Ranchi data) is derived from an instrument mounted on a 32-m tall tower at the Birla Institute of Technology Mesra in Ranchi, India (Dwivedi et al., 2014) with an average elevation of 609 m above sea level in a tropical region. The site has a few buildings in between east and northwest; agriculture land in between northwest and west; and residential area, and dense trees in between southeast and east. The site also has a relatively flat area in between southeast and west which is free from any obstacle (Srivastava and Sharan, 2015). A fast response sensor (CSAT3 Sonic Anemometer) at a height of 10 m with an average elevation 609 m above sea level provides the temperature and the three components of wind at a 10 Hz frequency. The eddy covariance technique (Stull 1988) is used to estimate heat and momentum fluxes at one-hour time resolution, however the hourly temperature at 2-m is determined by averaging temperature observations available at a temporal scale of 1 minute from the slow response sensors located at logarithmic heights on the same tower. We have utilized

hourly data for considered variables. Apart from this we have also utilized the ERA5-Land reanalysis dataset available at
$0.10^o \times 0.10^o$ spatial resolution to evaluate the spatial distribution of the model simulated near surface atmospheric variables.
For consistency, we have regridded the model output to the same grid resolution of reanalysis/observed dataset.

## 3 Numerical simulations

To analyze the impacts of newly installed similarity functions together with the existing functional forms in the surface layer
scheme of WRFv4.2.2, the performance of the default and newly installed similarity functions is investigated in two steps. The
first one is independent of the WRF model. Namely, we apply Eqn. (B8) (Appendix B) to iteratively determine $C_D$ and $C_H$ as
a function of $\zeta$ by prescribing the bulk Richardson number ($Ri_B$) and surface roughness parameters for momentum ($z_0$) and
heat ($z_h$). The value of $\zeta$ is estimated by calculating the root of least magnitude of Eqn. (B8) for a given value of $Ri_B$. Once $\zeta$
is calculated then utilizing it in Eqns. (B9) and (B10), the values of $C_D$ and $C_H$ can be estimated. We call this in the following
offline simulation. For the computation, z is taken as 10 m and $Ri_B$ is in the range $-2 \le Ri_B \le 0$. The offline simulations are
carried out over three different surface types by considering surface roughness ($z_0$) to be 0.01 m (smooth surface), 0.1 m
(transition surface) and 1 m (rough surface) to analyze the impact of roughness of underlying surface on the simulation of $\zeta$,
$C_D$ and $C_H$.
The second step is to apply all the parameterizations of the similarity functions in the WRF model version 4.2.2 over
an Indian land site whose output is compared then with the observations during the pre-monsoon (March-April-May; MAM)
season of the year 2009. The simulations have been conducted over a nested domain centred around the location of a
micrometeorological tower installed at Ranchi ($23.412^o$N, $85.44^o$E), India (Figure 3). Domain d01 ($\mathbf{6 \times 6}$ km$^2$) consists of 233
east-west and 210 north-south grid points and domain d02 ($\mathbf{2 \times 2}$ km$^2$) consists of 223 east-west and 196 north-south grid
points which covers $\mathbf{1398 \times 1260}$ km$^2$ and $\mathbf{446 \times 392}$ km$^2$ spatial area around the centre point, respectively. Each domain
was configured with 50 vertical eta levels from surface to top of the atmosphere. We kept five vertical levels below 100 m
height. Initial and boundary conditions were taken from ERA5 global atmospheric reanalysis dataset at a resolution of
$\mathbf{0.25^o \times 0.25^o}$ and boundary conditions were forced every 6 hours. For land use and land cover (LU/LC) information, we
have used dataset from MODIS (Moderate Resolution Imaging Spectroradiometer; Friedl et al., 2002). Various physical
parameterizations utilized in the simulations are listed in Appendix C. In this study, four sets of simulations were carried out,
as given in Table 1.
Note that the revised MM5 surface layer scheme has lower limits on the values of $\mathbf{u}_*(\mathbf{> 0.001}$ m s$^{-1}$) and $\mathbf{U(> 0.1}$
m s$^{-1}$) that allow nocturnal values of $\mathbf{u}_*$ at night and control $Ri_B$ values to be inordinately high, respectively (Jimenez et al.,
2012). However, the stability parameter $\zeta$ or $Ri_B$ is not restricted in the revised MM5 surface layer scheme, which gives
complete freedom to the WRF model to show its sensitivity to the tested similarity functions (Jimenez et al., 2012). Moreover,
some of the LES studies reported in the literature suggest that the friction velocity cannot be zero when the mean wind drops
to zero, i.e., there should be a minimum friction velocity that is proportional to the $\mathbf{w}_*$ (Schumann, 1980). For this purpose,
the existing version of the revised MM5 scheme sets $\mathbf{0.001}$ m s$^{-1}$ as the minimum value of $\mathbf{u}_*$ based on the recommendations
by Jimenez et al. (2012). Thus, to avoid the complexity that arises when mean wind drops to zero, the updated revised MM5
scheme proposed in the present study also utilizes a minimum value of $\mathbf{u}_*$ ($> \mathbf{0.001}$ m s$^{-1}$) as suggested by Jimenez et al.
(2012) in the existing version of the revised MM5 scheme. Moreover, the scheme uses constant values of $\mathbf{z_0}$, while the values
of $\mathbf{z_h}$ are calculated from the expression suggested by Brutsaert (1982).
The whole simulation period is divided into segments of 4 days with 24 h overlapping time between different
segments to ensure continuity. The model is initialized at 0000 UTC of the first day of each simulation and runs for 96 hours.
In order to avoid the potential spin-up problems at the beginning of the simulation, we discard the first day of each simulation
as spin up time and consider the last three days for the analysis (Jimenez et al., 2010; 2012).
For the evaluation of the real-case simulations, different statistical parameters such as mean absolute error (MAE),
root mean square error (RMSE), mean bias (MB), index of agreement (IOA), different measures of correlation coefficient
(CC), mean bias (%) (bias), and standard deviation of the model predicted output normalized by that of the observations are
used. A brief description of the performance indicators for validation utilized in the present study is given in Appendix C.
**4 Results**
**4.1 Offline simulations**
To analyze the functional dependence of $\zeta$, $C_D$ and $C_H$ on the utilized forms of similarity functions, the offline simulations
independent of the WRF model have been conducted utilizing newly installed functions (BD71, CL73, and KY90) together
with F96 functions existing in the default version of the surface layer scheme of the WRF model for three different roughness
lengths for momentum $(z_0)$, which are representative of smooth $(z_0 = 0.01 \text{ m})$, transition $(z_0 = 0.1 \text{ m})$, and rough
$(z_0 = 1.0 \text{ m})$ surfaces. The results for $\zeta$ (Figure 4a, b, and c) with $Ri_B$, $C_D$ (Figure 4d, e, and f) and $C_H$ (Figure 4g, h, and i)
with $\zeta$ across various surface types and sublayers have been analyzed. The different sublayers associated with convective
stratification include dynamic (DNS), dynamic-dynamic convective transition (DNS-DCS), dynamic convective (DCS),
dynamic convective-free convective transition (DCS-FCS), and free convective (FCS) (Srivastava and Sharan, 2021). Note
that the sublayers DNS $(-0.04 \leq \zeta \leq 0)$ and DNS-DCS transition $(-0.12 \leq \zeta < -0.04)$ are corresponding to weakly to
moderately unstable conditions, while sublayers DCS $(-1.20 \leq \zeta < -0.12)$, DCS-FCS $(-2.0 \leq \zeta < -1.20)$, and FCS $(\zeta <$
$-2.0)$ belong to moderately to strongly convective conditions (Srivastava and Sharan, 2015).
It is found that the simulated values of $\zeta$ at smaller values of $Ri_B$ (i.e., in DNS to DCS) from different forms of
similarity functions are found to be almost identical to the F96 functional forms (Figure 4a-c). Moreover, results from the
BD71, CL73, and F96 functions are even similar at higher instabilities (i.e., the whole range of $\zeta$ values), while they differ
strongly from values obtained using the KY90 functions (Figure 4a-c). Notably, BD71, CL73, and F96 functional forms predict
relatively smaller absolute values of $\zeta$ for a given value of $Ri_B$. However, KY90 functions are found to produce a relatively
larger magnitude of $\zeta$ for a given value of $Ri_B$. This behaviour is found to be consistent for all ratios $z/z_0$ (Figures 4a-c)
representative of smooth, transition, and rough surfaces. A relatively larger magnitude of $\zeta$ for a given value of $Ri_B$ and the
smaller values of $\psi_m$ and $\psi_h$ (Figure 2) in KY90 functional forms implies that the momentum and heat fluxes predicted using
KY90 functions will be smaller than those anticipated in BD71, CL73, and F96 functional forms.
Figure 4d-f shows the variation of $C_D$ with $\zeta$ estimated using BD71, CL73, KY90, and F96 functional forms over
different surfaces. Notice that the $C_D$ values calculated from BD71, CL73, and F96 forms of functions are relatively higher
than those produced by KY90 functional forms and continue to rise as instability progresses from DCS to FCS. It is important
to highlight that $C_D$ estimated using KY90 functions shows a non-monotonic behaviour, which is consistent with the observed
behaviour of $C_D$ over the Indian region reported in the literature (Srivastava and Sharan, 2019; 2021). Note that this non-
monotonic behaviour is consistent for all three cases of different roughness lengths (Figure 4d-f).
On the other hand, across all three surfaces, one can see that the values of $C_H$ estimated from all four functional forms
increase with increasing instability (Figure 4g-i), while the rate of increase of $C_H$ in KY90 functions is relatively slower.
Moreover, BD71, CL73, and F96 functions predict almost similar values over all three types of surfaces. Noticeably, $C_H$
estimated using KY90 functions also exhibits non-monotonic behaviour with $\zeta$ over rough surfaces, which contradicts the
predictions of the other three functional forms. In addition, it is important to note that $C_D$ and $C_H$ predicted by KY90 functional
forms are found to bound by twice their near-neutral values, while the other functional forms predict continuously increasing
values of $C_D$ and $C_H$ on increasing instability.
Note that the error caused by different values of $z_0$ can be so large that the stability dependence of using different
forms of similarity functions is less important in the computation of $C_D$ and $C_H$. As a result, three different values of $z_0$ have
been chosen, similar to a recent study by Srivastava and Sharan (2021), which are representative of smooth ($z_0 = 0.01$ m),
transition ($z_0 = 0.1$ m), and rough ($z_0 = 1.0$ m) surfaces to account for the impacts of using different $z_0$ on the estimation of
$C_D$ and $C_H$ from different functional forms of similarity functions in offline simulations.
Moreover, Figure 4 depicts the offline simulations with equal values of $z_0$ and $z_h$. While in the revised MM5 surface
layer scheme available in the WRF model, the values of $z_0$ and $z_h$ are not the same. Thus, we have also attempted to discuss
the results from the offline simulations with different values of $z_h$, assuming $z_0 = 0.1$ m. Figure S2 (supplementary material)
shows the variation of $\zeta$ with $Ri_B$, $C_D$, and $C_H$ with $\zeta$ calculated from the bulk flux algorithm using similarity functions
corresponding to BD71, CL73, KY90, and F96 with different values of $z_h$ while $z_0$ is fixed. The values of $z_h$ are taken such
that the ratio $\ln(z_0/z_h)$ assumes 0.1, 1, 2, 3, and 4. Figure S2 clearly shows that the estimated values of $\zeta$ are similar in near-
neutral to moderately unstable conditions for all values of $z_h$; however, relatively smaller values have been found as the ratio
$\ln(z_0/z_h)$ increases for each form of similarity function. Since the computation of $C_D$ does not involve the values of $z_h$ (Eqn.
B9), the estimated values of $C_D$ for each form of similarity function are found to be approximately the same for different values
of $z_h$. However, in the case of $C_H$, differences are clearly visible if one uses different values of $z_h$. The estimated $C_H$ using
various similarity functions behaves similarly for different values of $z_h$, while the magnitude decreases as the ratio $\ln(z_0/z_h)$
increases.
Hence, it is evident that the BD71, CL73, and F96 functional forms predict values of $\zeta$, $C_D$, and $C_H$ that are almost
same over all the three different surface types. However, using KY90 functions compared to other commonly used $\varphi_m$ and
$\varphi_h$, one can expect a significant reduction in the estimated values of transfer coefficients in moderately to strongly unstable
stratification.
**4.2 Results of the WRF model using different sets of integrated similarity functions**
In this section, observational and reanalysis datasets have been used to analyze the simulations performed with WRFv4.2.2
utilizing newly installed and default $\varphi_m$ and $\varphi_h$. The model simulated output has been extracted at the location of the flux
tower and compared against the observations derived from the flux tower installed at Ranchi (23.412N, 85.440E), India. The
mean spatial patterns of certain variables averaged over daytime (04:00-12:00 UTC) have been compared against the ERA5-
Land reanalysis dataset. Further, to access the effects of newly installed functions under free convective conditions, the mean
spatial patterns of considered variables averaged across strong convective conditions (hours in which $\zeta < -10$ over most of
the domain) have been analyzed against respective hours of ERA5-Land reanalysis data. Bilinear interpolation has been used
to interpolate the model output to the same grid resolution as the ERA5-Land data in order to allow a consistent comparison.
**4.2.1 Evaluation against observations derived from flux tower installed at Ranchi (India)**
Figure 5 depicts the variation of (a) $\zeta$ with $Ri_B$, (b) $C_D$, and (c) $C_H$ at the first model level with $\zeta$ from different experiments
(Exp1, Exp2, and Exp3) and CTRL simulation. Although the absolute values of the parameters differ from each other due to
the different prescribed roughnesses, the variation of $\zeta$ with $Ri_B$, $C_D$ and $C_H$ with $\zeta$ is very similar to the offline results. Note
that, at the moment, due to the inaccessibility of long-term data on detailed surface properties such as vegetation structure
needed to quantify the roughness length, we do not have an access to the precise values of $z_0$ and $z_h$ at the Ranchi station.
Moreover, the values of $z_0$ and $z_h$ do not directly involve in the estimation of $C_D$, $C_H$, and the surface fluxes from the
observational data, while they are important in computing these variables using the MOST framework. Thus, the default value
of $z_0$ is used in the revised MM5 surface layer scheme available in the WRF model, which is found to be approximately in the
range $0.1 - 0.2$ m at the Ranchi station. We wish to highlight that the $z_0$ used in the WRF model simulations at the Ranchi
station is nearly similar to the case of $z_0 = 0.1$ m presented in Figure 4, and the offline simulations also indicate that the
behaviour of the estimated $C_D$ and $C_H$ with $\zeta$ remains almost the same for different values of $z_0$ with slightly varying
magnitudes. Thus, one can interpret the results of $C_D$ and $C_H$ shown in Figures 4 and 5 from the offline simulations and the
WRF model, respectively, and can compare the WRF model simulated $C_D$ with the observed one at the Ranchi station.
Although the model simulations and observed data may have a different $z_0$, the comparison of model simulated variables with
the Ranchi data allows for an impression of the structural behaviour of model results as a function of stratification compared
with measurements.
It is clear from Figure 5 that the values of simulated variables are found to be almost identical in DNS to DCS
sublayers for all the experiments. Moreover, in FCS, the results obtained from Exp1, 2 and CTRL simulation are found to be
nearly similar however, relatively strong differences have been found in results from Exp 3 (Figure 5a, b, and c). Simulated $\zeta$
for a given $Ri_B$ in Exp2 and CTRL simulation are similar and found to be relatively smaller in magnitude than Exp1 and Exp3
in FCS. However, the absolute values of $\zeta$ in Exp3 (KY90 functions) are relatively higher in FCS than in all other experiments.
Figure 5b shows the variation of simulated $C_D$ with $\zeta$ from different experiments. Purple circles denote the variation
of observed $C_D$ with $\zeta$ at the location of flux tower (Figure 5b). It is found that the observed $C_D$ increases as the instability
increases from DNS to DCS and has the maximum value in the DCS (at $\zeta = -0.1$ approx.) and then starts to decrease as
instability grows further from DCS to FCS. It is evident that $C_D$ simulated using $\varphi_m$ and $\varphi_h$ based on class 4 functions (Exp3)
exhibits non-monotonic behaviour (Figure 5b), which is consistent with the observed behaviour of $C_D$ (Srivastava and Sharan,
2015; 2021). The magnitude of $C_D$ predicted in Exp3 is significantly smaller than that simulated from other experiments as
well as CTRL simulation, specifically in FCS. This may be due to the large differences between the KY90 functional forms of
$\psi_m$ and $\psi_h$ and other forms of functions. On the other hand, $C_D$ simulated using $\varphi_m$ and $\varphi_h$ based on the first three classes
(Exp1, Exp2, and CTRL simulation) increases continuously as instability grows from DNS to FCS (Figure 5b).
However, it is found that the $C_D$ predicted from the original forms of class 4 functions (Exp3) show large disagreement
with its observed behaviour, as the predicted $C_D$ starts decreasing at $\zeta$ lying in FCS, which is different from that observed, i.e.,
$\zeta$ lying in DCS. As a result, the study also highlighted the necessity of fine-tuning the original KY90 functional forms and
evaluating their performance in the WRF model with additional observational datasets from various land sites and seasons.
Note that Srivastava and Sharan (2021) tuned the original forms of class 4 functions by enforcing the matching of the
point at which both observed and model predicted $C_D$ attain their maximum value. However, more studies in terms of predicting
the observed variation of the non-dimensional vertical gradients of mean wind speed and temperature with $\zeta$ are essential to
further tune the original KY90 functions for the Indian region using observed data from various locations under different
seasons.
Further, we would like to point out that currently no observational datasets are available which show a better
agreement with the KY90 functions over Indian land. However, it is desirable to further validate these functional forms over
Indian land once such observational datasets become available.
We wish to highlight that utilizing KY90 (Exp3) functions in the revised MM5 scheme of the WRF model makes it
consistent in predicting $C_D$ with its observed non-monotonic behaviour over the Indian region.
The variation of simulated $C_H$ with $\zeta$ from different experiments is shown in Figure 5c. $C_H$ simulated from Exp1-3 as
well as CTRL simulation shows continuously increasing behaviour with $\zeta$. The magnitude of simulated $C_H$ from CTRL
simulation and Exp1-2 is relatively higher than that of Exp3 in FCS beyond $\zeta < -10$ (approximately). It is also evident that
at higher instabilities, even $C_H$ shows non-monotonic behaviour with $\zeta$ (Figure 5c). We wish to point out that a relatively larger
scatter has been found in the values of $C_H$ than $C_D$. As the WRF model utilizes constant values for $z_0$, while $z_h$ is calculated
using expression suggested by Brutsaert (1982). The relatively large scatter in the values of $C_H$ simulated from the WRF model
can be due to the parameterization of the ratio of momentum and scalar roughness lengths in the model.

331   Note that the transfer coefficients $C_D$ and $C_H$ shown in Figure 5 are at the reference height corresponding to the lowest

model grid level, which is ~12 m in the present study. However, we have also analyzed the behaviour of $C_D$ and $C_H$ at 10 m
height with $\zeta$ and found that they behave similarly to those presented in Figure 5.

334   The analysis presented here indicates that the KY90 functions in the revised MM5 surface layer scheme are found to

be appropriate in producing non-monotonic behaviour of $C_D$ consistent with its observed nature. However, all other functional
forms of $\varphi_m$ and $\varphi_h$ produce $C_D$, which increases continuously with $\zeta$ from DNS to FCS.

337   To quantify the uncertainties involved in the simulated surface fluxes and certain near-surface variables using KY90

(Exp3) as well as other functional forms (Exp1-2 and CTRL simulation), model simulations have been compared against the
observations. Figure 6 compares the model-simulated (a) $u_*^2$ (m$^2$ s$^{-2}$) (representative of momentum flux), (b) SHF (W m$^{-2}$)
(sensible heat flux), (c) $U_{10}$ (m s$^{-1}$) (10-m wind speed), and (d) $T_{2m}$ (K) (2-m temperature) with the observed data obtained
from the flux tower at Ranchi (23.412N, 85.440E), India. The model output was extracted at a single grid point closest to the
flux tower to allow a consistent comparison. In Figure 7, a Taylor diagram is displayed along with the normalized standard
deviations and correlations of considered variables. Figure 8 shows the scatter plot between CC vs. RMSE for considered
variables simulated using different experiments. In case of $u_*^2$, Exp1 and Exp2 are found to be comparable to the CTRL
simulation, while Exp3 considerably improved the simulation of $u_*^2$ (Figures 6a, 7 and 8). Exp3 reduced MAE (RMSE) from
0.09 (0.16) m$^2$ s$^{-2}$ to 0.08 (0.14) m$^2$ s$^{-2}$ (Table 2; Figures 7 and 8) and improved the CC (0.74) and IOA (0.84) for $u_*^2$ (Table 2).
A Q-Q plot is shown in Figure S3a (supplementary material) suggesting that Exp3 (KY90 functions) is found to be slightly
better than all other experiments and CTRL simulation for $u_*^2$. For SHF, all the experiments are comparable to the CTRL
simulation; however, Exp3 shows less scatter than other experiments (Figure 6a).

350   In case of $U_{10}$, Exp3 shows less scatter and appears to be closer to the observations than other experiments (Figure

6c). Exp3 noticeably improved the simulation of $U_{10}$ by reducing MAE (RMSE) from 1.20 (1.54) m s$^{-2}$ to 1.16 (1.47) m s$^{-2}$ and
MB up to 5 % (Figures 6c, and 7; Table 2). It considerably improved the CC (IOA) for $U_{10}$ from 0.66 (0.73) to 0.68 (0.75)
(Figure 7 and Table 2). A Q-Q plot (Figure S3b: supplementary material) reveals that Exp3 is found to be better than all other
experiments and CTRL simulation for $U_{10}$. Thus, the KY90 functions in the surface layer scheme of the WRF model
considerably improve the model in simulating $U_{10}$ (Figures. 6c, 7, 8, and S2b) at the location of the flux tower. Further, in case
of $T_{2m}$, Figures 7 and 8 exhibit that all the experiments are found to be comparable with the CTRL simulation.

357   Note that earlier studies, especially the ones done in the GABLS model intercomparison projects, have studied the

impacts of the similarity functions on the modelled profiles and fluxes (though mostly for stable conditions). However, they
learnt that applying different stability functions in the surface and boundary layer parameterizations may trigger unnatural
kinks in the model simulated wind speed and temperature profiles. Here, we have analyzed the profiles of $U_{10}$ and $T_{2m}$ simulated
from WRF model using different similarity functions in the surface layer scheme for the occurrence of unnatural kinks in their
values. One can see that the $U_{10}$ predicted from CTRL simulation, as well as different experiments corresponding to different
similarity functions at certain hours goes higher than that of its observed maximum value (approx. 8 m s$^{-1}$) (Figure S4:
supplementary material). These relatively higher magnitudes may be linked with some localised weather phenomenon
characterized by rapid changes in weather including strong wind, lightning and thunderstorms and are justifiable. However,
the simulated $T_{2m}$ from different similarity functions are found to be in line with the observed values across the whole
simulation period (Figure S5: supplementary material). This suggests that the values of $U_{10}$ and $T_{2m}$ predicted from WRF
model are found to be in justifiable range and no unnatural kinks have been found.
**4.2.2 Evaluation of mean spatial distribution of simulated variables against ERA5-Land reanalysis data during daytime**
In this section, mean spatial distribution of simulated variables from different experiments as well as CTRL simulation
averaged during daytime (04:00-12:00 UTC) for entire simulation period, is compared with the ERA5-Land reanalysis data.
Figure 9 depicts the mean spatial patterns of simulated $\zeta \left( = \frac{z}{L} \right)$ (a1 − 4), $C_D$ (c1-c4), and $C_H$ (e1-4) from CTRL simulation
and other experiments, as well as their differences with respect to CTRL simulation. It is found that the absolute value of $\zeta$
simulated in Exp3 (KY90 functions) is relatively smaller than CTRL simulation (Figure 9b3) across the whole domain, which
is consistent with Figure 5a, and offline simulations presented in Figure 4(a-c). This could be because the magnitude of KY90
functions ($\varphi_m$ and $\varphi_h$) (Figure S1: supplementary material) is relatively smaller than the functions employed in default scheme
(CTRL simulation).
On the other hand, Exp1 also provides slightly smaller absolute values of $\zeta$ (Figure 9b1), while Exp2 is almost
comparable to the CTRL simulation (Figure 9b2). Model simulated $C_D$ is found to be relatively smaller in Exp3 than CTRL
simulation (Figure 9d3), while Exp1 and Exp2 provide comparable values of $C_D$ to CTRL simulation (Figure 9d1-2). In the
case of $C_H$, the simulated values from different experiments are comparable to the CTRL simulation over the whole study
domain (Figure 9f1-3). Note that simulated $C_H$ is found to be comparable in all the experiments while one can see slight
differences in $C_D$ in Exp3 than all other experiments which may be related to the fact that only $\varphi_m$ functions are involved in
the computation of $C_D$ (Eqn. 1), and the differences between $\varphi_m$ corresponding to Exp3 are relatively more than $\varphi_h$, so are
the differences in $C_D$. The hatched regions in Figure 9 shows the differences between simulated variables from different
experiments with respect to CTRL simulation are statistically significant at 95% confidence level.
The slight differences in $C_D$ in Exp3 reflected further in the simulated $u_*^2$ m$^2$ s$^{-2}$ (a measure of momentum flux) (Figure
10b3). A slight reduction has been found in simulated $u_*^2$ in Exp3 compared to the CTRL simulation over some parts of the
domain (Figure 10b3), while in Exp1 and Exp2 values are comparable with the CTRL simulation (Figure 10b1-2). In case of
SHF and LHF, the mean spatial distribution from all the experiments is found to be consistent with the ERA5-Land reanalysis

data, and the magnitude of differences between model simulation and ERA5-Land data is comparable for all the experiments (Table S1: supplementary material).

For $T_{2m}$ (upper panel of Figure 11), $T_S$ (middle panel of Figure 11), and $U_{10}$ (lower panel of Figure 11), mean spatial distribution from different experiments and CTRL simulation agreed well with slightly varying magnitude to the ERA5-Land reanalysis data. One can see a warm bias up to 2 K (3 K) for $T_{2m}$ ($T_S$) simulated from different experiments and CTRL simulation over most of the domain. For $T_{2m}$, bias, RMSE, and PCC between different experiments together with CTRL simulation and ERA5-Land reanalysis data are found to be comparable (Table S1: supplementary material). However, Exp3 slightly improved the PCC from 0.50 to 0.51 for $T_S$ (Table S1: supplementary material). Further, in the case of $U_{10}$, all the simulations exhibit overprediction over the whole domain (lower panel of Figure 11: b1-4) and Exp3 is found to be slightly better than all other experiments as well as CTRL simulation as it reduced bias% (RMSE) from 32.28 (0.54) m s$^{-2}$ to 32.06 (0.53) m s$^{-2}$ and improved the PCC from 0.89 to 0.91 (Table S1: supplementary material).

### 4.2.3 Evaluation of newly installed functions during strong unstable conditions with respect to ERA5-Land reanalysis data

This section describes the impacts of utilizing different similarity functions ($\varphi_m$ and $\varphi_h$) on simulated variables during highly convective regime (i.e., $\zeta < -10$) with respect to the ERA5-Land reanalysis dataset. Since the functional forms of $\psi_m$ and $\psi_h$ are almost identical in near-neutral to moderately unstable conditions, however, in strong unstable conditions, the differences between different functional forms are more pronounced. Thus, the corresponding differences in the simulated values of considered variables are expected to be more pronounced during highly convective regimes. For this purpose, the model output has been extracted for those hours in daytime which show $\zeta$ smaller than $-10$ over most of the domain and compared with the respective hours of ERA5-Land reanalysis data.

Figure S6 (Supplementary material) depicts the mean spatial distribution of $\zeta$ (a1-4), $C_D$ (c1-4), and $C_H$ (e1-4) as well as their deviations from CTRL simulation. Notice that the magnitude of differences for all variables ($\zeta$, $C_D$, and $C_H$) in this case are found to be larger than the case of mean spatial patterns averaged during the whole daytime (section 6.2.2). It is evident from Figure S6b3 (supplementary material) that Exp3 produce large absolute values of $\zeta$ and smaller values of $C_D$ and $C_H$ (Figures S6b3, d3 and f3: supplementary material) than all other experiments and the CTRL simulation. While Exp1 and Exp2 are found to be comparable to the CTRL simulation for both $C_D$ and $C_H$ (Figures S6d1-2 and f1-2).

The model simulations for $T_{2m}$ and $T_S$ do not capture the spatial patterns well in comparison to ERA5-Land data (Figures S7a1-5 and S8a1-5: supplementary material). All experiments, as well as the CTRL simulation, exhibit overprediction across the whole domain (Figures S7b1-4 and S8b1-4). We wish to highlight that the differences between various experiments and CTRL simulation are seen up to 0.5 K for $T_{2m}$ (Figure S7c1-3: supplementary material) as well as $T_S$ (Figure S8c1-3) which is slightly higher than the case of mean spatial patterns averaged over whole daytime (upper and middle panels of Figure 11). For $T_{2m}$, it is evident from Figure S7 (supplementary material) and Table 3 that Exp3 noticeably reduced the bias%

(RMSE) from 0.64 (2.13) K to 0.62 (2.10) K and improved the PCC from 0.43 to 0.46 (approximately 6%). In case of $T_S$ as
well, Exp3 slightly improved the PCC and reduced the bias% (RMSE) from 1.25 (4.01) K to 1.24 (3.97) K (Table 3 and Figure

12).

For $U_{10}$, the mean spatial patterns simulated using different experiments agreed well with the ERA5-Land reanalysis

data (Figure S9a1-5: supplementary material) and the magnitude of biases is found to be up to 1 m s$^{-1}$. Exp3 outperformed all
other experiments and the CTRL simulation by lowering the bias% from -4.96 to -0.28 m s$^{-1}$ and improved the PCC from 0.34
to 0.36 with comparable RMSE values (Figures S9 and 12; Table 3).

The results presented so far suggest that the changes corresponding to different functional forms of similarity

functions in the surface layer parameterization of the WRF model are more pronounced in convective conditions during
daytime hours. For the number of grid points over the study domain that are being affected by the changed similarity functions,
no fixed pattern was found; however, the changes depend on the considered variable and similarity functions. Furthermore,
we observe that the changes are more pronounced in grids that experience strong instability during the daytime.

## 5 Summary and concluding remarks

In the present study, the revised MM5 surface layer scheme of the WRFv4.2.2 model has been modified to incorporate $\varphi_m$
and $\varphi_h$ suggested by Kader and Yaglom (1990) to make it consistent in producing the transfer coefficient for momentum ($C_D$)
in line with its observed behaviour. The revised MM5 scheme is modified in such a way that it contains all commonly used
$\varphi_m$ and $\varphi_h$ under convective conditions instead of a single functional form. Various alternatives of $\varphi_m$ and $\varphi_h$ in the modified
scheme can be controlled by a flag (psimhu_opt) that has been introduced in the physics section of the namelist file. The
impacts of utilizing different functional forms of $\varphi_m$ and $\varphi_h$ in the proposed scheme have been evaluated using offline
simulations (with bulk flux algorithm) as well as real-case simulations with WRFv4.2.2 model. The model-simulated surface
turbulent fluxes and certain near-surface variables have been compared with observational data from a flux tower at Ranchi
(23.412N, 85.440E; India), and the spatial patterns have been evaluated with the ERA5-Land reanalysis dataset.

Offline simulations indicate that at nearly neutral to moderately unstable conditions, $\zeta$ simulated using various

functional forms of $\varphi_m$ and $\varphi_h$ is comparable, and as the instability grows (free convective conditions), the differences
between different experiments become more pronounced. This might be connected to the corresponding variations between
different functional forms of similarity functions in the respective regimes. Similarly, for simulated $C_D$, Exp3 (KY90 functions)
demonstrates nonmonotonic behaviour with $-\zeta$ across all three surface types (representing smooth, transition, and rough
surfaces), which is consistent with its observed behaviour. However, all other experiments and CTRL simulation indicate
continuously increasing $C_D$ with $-\zeta$ from near-neutral to free convective conditions over all three surface types, which is
inconsistent with its observed behaviour over the study domain. The non-monotonic behaviour of $C_D$ in Exp3 (KY90 functions)
may be associated to the analogous non-monotonic behaviour of the corresponding $\psi_m$ in the respective regime.
In real-case simulations, the model simulated $\zeta$, $C_D$ and $C_H$ are found to be consistent with the offline simulations.
One can see that the variation of $C_D$ in Exp3 (KY90 functions) with $-\zeta$ is nonmonotonic, as reported in offline simulations
and found to be consistent with its observed behaviour. This indicates that the KY90 functions in the surface layer scheme of
the WRF model make it compatible in producing $C_D$ consistent with its observed behaviour over Indian land. As compared
with the observations over Ranchi (India), the simulations using KY90 (Exp3) functions are found to perform better for most
of the considered variables compared to all other experiments. Further, in the mean spatial distribution averaged during daytime
(04:00–12:00 UTC) over the entire simulation period, the significant increase in absolute value of $\zeta$ from Exp3 resulted in a
noticeable reduction in the values of $C_D$ and $C_H$, which further impacted the simulated values of $T_S$, $T_{2m}$, and $U_{10}$. When
compared with the ERA5-Land reanalysis data, the spatial patterns for $T_{2m}$, $T_S$, and $U_{10}$ from Exp3 (KY90 functions) provided
more consistent results. A reduction has been found in bias (%) and RMSE values for $T_S$, and $U_{10}$. Moreover, in case of highly
convective regime ($\zeta < -10$), Exp3 (KY90 functions) slightly improved the performance of the model by reducing the bias
(%) and RMSE for $T_{2m}$, $T_S$, and $U_{10}$ and increasing the correlation to some extent.
Thus, it is concluded that the similarity functions proposed by Kader and Yaglom (1990) (KY90 functions; Exp3) are
found to be more appropriate for use in the WRF model as they can simulate $C_D$ consistent with its observed behaviour and
improve the simulation for most of the considered variables over the study domain. However, due to the limited spatial
coverage of the domain considered in this study and the limited availability of observational data, KY90 functional forms need
to be further evaluated in the WRF modeling framework utilizing observations from other sites. The modified surface layer
scheme proposed in this study could enhance the potential applicability of the WRF modeling framework for the community
in investigating the role of different functional forms of similarity functions under convective conditions for selected
events/case studies such as extreme weather events, heat waves during summer, cyclonic storms, and fog predictions using the
WRF model.
**Appendix A**
Here, the detailed description of commonly used functions ($\varphi_m$ and $\varphi_h$) in numerical models under convective conditons is
provided.
Based on Businger (1966) and A. J. Dyer [1965, unpublished work; see Businger (1988) for details] the expressions
for $\varphi_m$ and $\varphi_h$ are as follows:
$\varphi_m = (1 - \gamma_m \zeta)^{-\frac{1}{4}}$                    (A1)
$\varphi_h = Pr_t (1 - \gamma_h \zeta)^{-\frac{1}{2}}$                    (A2)
in which $\gamma_m = 15$, $\gamma_h = 9$, and $Pr_t = 0.74$ is the turbulent Prandtl number. Note that in case of Dyer (1974) the values of
$\gamma_m = \gamma_h = 16$ and $Pr_t = 1.0$. These functions commonly known as Businger-Dyer similarity (BD) functions and do not
satisfy the classical free convection limit (Srivastava et al. 2021).

The similarity functions proposed by Carl et al. (1973) under convective conditions are applicable for the range
$-10 \leq \zeta \leq 0$. The expressions for $\varphi_m$ and $\varphi_h$ suggested by Carl et al. (1973) are:
$\varphi_m = (1 - \beta_m \zeta)^{-\frac{1}{3}}$                                                          (A3)
$\varphi_h = (1 - \beta_h \zeta)^{-\frac{1}{3}}$                                                          (A4)
in which $\beta_m = \beta_h = 15$. However, based on various studies reported in the literature $\beta_m$ and $\beta_h$ can take different values. For
example, Delage and Girard (1992) proposed $\beta_m = \beta_h = 40$, on the other hand, Fairall et al. (1996) suggested that $\beta_m = \beta_h = $
12.87.

Fairall et al. (1996, 2003) proposed an interpolation function applicable for the entire range of atmospheric instability,
which was based on BD functions and functions suggested by Carl et al. (1973). This interpolation function does not have the
gradient form ($\varphi_m$ and $\varphi_h$), as they have interpolated the integrated forms of the functions. We wish to highlight that the
revised MM5 surface layer scheme of Weather Research and Forecasting Model version 4.2.2 utilized the interpolation
functions suggested by Fairall et al. (1996).

Kader and Yaglom (1990) proposed a three-sublayer model under convective conditions. The dynamic sublayer
corresponds to near-neutral conditions in which $\varphi_m = 1$ and $\varphi_h = \mathrm{Pr_t}$. Further, in the dynamic convective sublayer,
mechanical energy is in the x direction, while buoyancy-induced energy is in the z direction. Thus, in this sublayer, the
functional forms for similarity functions, as determined by dimensional analysis, are
$\varphi_m(\zeta) = A_u (-\zeta)^{-\frac{1}{3}}$                                                       (A5)
$\varphi_h(\zeta) = A_T (-\zeta)^{-\frac{1}{3}}$                                                       (A6)
in which $A_u$ and $A_T$ are constants.

Moreover, in the free-convective sublayer, buoyancy dominates the mechanical production of energy, and the
pressure redistribution term feeds the buoyant energy in the vertical direction into the horizontal direction (Kader and Yaglom,
1990). Thus, in this case, the dimensional analysis suggests
$\varphi_m(\zeta) = B_u (-\zeta)^{\frac{1}{3}}$                                                       (A7)
$\varphi_h(\zeta) = B_T (-\zeta)^{-\frac{1}{3}}$                                                      (A8)
in which $B_u$ and $B_T$ are constants.

Thus, under unstable conditions, $\varphi_m$ exhibits a nonmonotonic behaviour with respect to $-\zeta$ as the three sublayer
theory suggested that for sufficiently large values of $-\zeta$, $\varphi_m$ varies as the $+1/3$ power of $\zeta$, in contrast to the case of the free
convection limit, where both $\varphi_m$ and $\varphi_h$ follow the $-1/3$ power law. In the literature, various expressions for $\varphi_m$ and $\varphi_h$ are
available based on the Kader and Yaglom (1990) three-sublayer model. However, the present study employs $\varphi_m$ and $\varphi_h$ based
on the expressions implemented in the surface layer scheme (CLM4.0) of NCAR-CAM5 (Zeng et al., 1998) model. The
expressions for $\varphi_m$ and $\varphi_h$ utilized in this study are as follows:
$\quad \varphi_{\mathrm{m}} = \begin{cases} (1 - 16\zeta)^{-\frac{1}{4}}, & -1.574 \leq \zeta \leq 0 \\ 0.7k^{\frac{2}{3}}(-\zeta)^{\frac{1}{3}}, & \zeta \leq -1.574 \end{cases}$ $\qquad\qquad$ (A9)
$\quad$ and
$\quad \varphi_{\mathrm{h}} = \begin{cases} (1 - 16\zeta)^{-\frac{1}{2}}, & -0.465 \leq \zeta \leq 0 \\ 0.9k^{\frac{4}{3}}(-\zeta)^{-\frac{1}{3}}, & \zeta \leq -0.465 \end{cases}$ $\qquad\qquad$ (A10)

$\quad$ Srivastava and Sharan (2021) classified these commonly used similarity functions stated above into four different classes based
$\quad$ on the exponents appearing in the expressions of $\varphi_{\mathrm{m}}$ and $\varphi_{\mathrm{h}}$. The classification is as follows:

$\quad$ **Class 1.** This class consists of functions having the exponents of $\varphi_{\mathrm{m}}$ and $\varphi_{\mathrm{h}}$ as $-1/4$ and $-1/2$ (as in Eqns. A1 and A2),
$\quad$ respectively from near-neutral to strong unstable conditions. $\varphi_{\mathrm{m}}$ and $\varphi_{\mathrm{h}}$ proposed by Businger (1971) and Hogstrom (1996)
$\quad$ are the examples of class 1 functions.

$\quad$ **Class 2.** In this class, the similarity functions ($\varphi_{\mathrm{m}}$ and $\varphi_{\mathrm{h}}$) having exponents of $\varphi_{\mathrm{m}}$ and $\varphi_{\mathrm{h}}$ as $-1/3$ for the entire range from
$\quad$ near-neutral to moderately unstable conditions (as in Eqns. A3 and A4), respectively are included. The functional forms
$\quad$ suggested by Carl et al. (1973) are the example of class 2 functions.

$\quad$ **Class 3.** $\varphi_{\mathrm{m}}$ and $\varphi_{\mathrm{h}}$ having exponents as $-1/4$ and $-1/2$, respectively in near-neutral conditions while $-1/3$ in strong
$\quad$ unstable conditions are included in this class. $\varphi_{\mathrm{m}}$ and $\varphi_{\mathrm{h}}$ based on Fairall et al. (1996), Grachev et al. (2000) and Fairall et al.
$\quad$ 2003 are some examples of class 3 functions.

$\quad$ **Class 4.** Functional forms of $\varphi_{\mathrm{m}}$ and $\varphi_{\mathrm{h}}$ having the exponents as $-1/4$ and $-1/2$, respectively in near-neutral conditions
$\quad$ however, $1/3$ for $\varphi_{\mathrm{m}}$ and $-1/3$ for $\varphi_{\mathrm{h}}$ in strong unstable conditions are classified in this class (as in Eqns. A9 and A10). The
$\quad$ three-sublayer model for $\varphi_{\mathrm{m}}$ and $\varphi_{\mathrm{h}}$ suggested by Kader and Yaglom (1990) (Zeng et al. 1998) is one of the examples of
$\quad$ functions in this class.
$\quad$ **Appendix B**
$\quad$ This section consists of a brief description of the computation of surface turbulent fluxes in the revised MM5 surface layer
$\quad$ scheme. In a homogeneous surface layer, the dimensionless wind and temperature gradients are defined as
$\quad \dfrac{kz}{u_*}\dfrac{\partial U}{\partial z} = \varphi_{\mathrm{m}}(\zeta),$ $\qquad\qquad$ (B1)
$\quad \dfrac{kz}{\theta_*}\dfrac{\partial \theta}{\partial z} = \varphi_{\mathrm{h}}(\zeta).$ $\qquad\qquad$ (B2)
where $L$ denotes the Obukhov length scale and $U$ is the wind speed at height $z$; $k$ represents the von Karman constant and its
value is taken as 0.4. Integrating Eqns. (B1) and (B2) with respect to $z$ leads to
$$U = \frac{u_*}{k}\left[\ln\left(\frac{z}{z_0}\right) - \left\{\psi_m(\zeta) - \psi_m\left(\frac{z_0}{L}\right)\right\}\right], \tag{B3}$$
$$(\theta_a - \theta_g) = \frac{\theta_*}{k}\left[\ln\left(\frac{z}{z_h}\right) - \left\{\psi_h(\zeta) - \psi_h\left(\frac{z_h}{L}\right)\right\}\right] \tag{B4}$$
in which $\psi_m$ and $\psi_h$ denote the integrated form of similarity functions $\varphi_m$ and $\varphi_h$. The roughness lengths for momentum and
heat are denoted by $z_0$ and $z_h$, respectively. The ground and surface air potential temperature are denoted by $\theta_g$ and $\theta_a$,
respectively. $\zeta(= \frac{z}{L})$ is the stability parameter and is defined as
$$\zeta = \frac{kgz}{\theta_a}\frac{\theta_*}{u_*^2} \tag{B5}$$
$\psi_m$ and $\psi_h$ can be calculated from the following expression (e.g., Panofsky, 1963):
$$\psi_m(\zeta) = \psi_h(\zeta) = \int_0^\zeta \frac{1 - \varphi_{m,h,q}(\zeta')}{\zeta'} d\zeta' \tag{B6}$$
The bulk Richardson number $(Ri_B)$ is given by:
$$Ri_B = \frac{g}{\bar{\theta}}\frac{(\theta_a - \theta_g)(z - z_0)^2}{U^2(z - z_h)} \tag{B7}$$
Substituting the values of U and $(\theta_a - \theta_g)$ from Eqns. (B3) and (B4) in Eqn. (B7), one gets
$$Ri_B = \zeta \left[\frac{\left(1 - \frac{z_0}{z}\right)^2}{\left(1 - \frac{z_h}{z}\right)}\right] \frac{\left[\ln\left(\frac{z}{z_h}\right) - \left\{\psi_h(\zeta) - \psi_h\left(\zeta\frac{z_h}{z}\right)\right\}\right]}{\left[\ln\left(\frac{z}{z_0}\right) - \left\{\psi_m(\zeta) - \psi_m\left(\zeta\frac{z_0}{z}\right)\right\}\right]^2} \tag{B8}$$
Note that Eqn. (B8) is a transcendental equation, and for a given value of $Ri_B$, the corresponding $\zeta$ value can be calculated
using any iterative method.
The bulk transfer coefficient for momentum ($C_D$) and heat ($C_H$) are defined as:
$$C_D = k^2\left[\ln\left(\frac{z + z_0}{z_0}\right) - \left\{\psi_m\left(\frac{z + z_0}{L}\right) - \psi_m\left(\frac{z_0}{L}\right)\right\}\right]^{-2} \tag{B9}$$
$$C_H = k^2\left[\ln\left(\frac{z + z_0}{z_0}\right) - \left\{\psi_m\left(\frac{z + z_0}{L}\right) - \psi_m\left(\frac{z_0}{L}\right)\right\}\right]^{-1}\left[\ln\left(\frac{z + z_h}{z_h}\right) - \left\{\psi_h\left(\frac{z + z_h}{L}\right) - \psi_h\left(\frac{z_h}{L}\right)\right\}\right]^{-1} \tag{B10}$$
Once we get $C_D$ and $C_H$, then the momentum ($\tau$), and sensible heat (H) fluxes are calculated using the following expressions:
$$\tau = \rho C_D U^2 \tag{B11}$$
$$H = -\rho c_p C_H U(\theta_a - \theta_g), \tag{B12}$$

**Appendix C**

In this section, the details of various physical parameterizations utilized in the real-case simulations using WRFv4.2.2 model and the different statistical indicators used for model evaluation.

The real-case simulations with the WRFv4.2.2 model utilised the Purdue Lin microphysics scheme (Lin et al., 1983); YSU (Hong, Noh, and Dudhia, 2006) PBL scheme; Kain-Fritsch (Kain and John, 2004) cumulus scheme; Dudhia (Dudhia, 1989) shortwave scheme; RRTM (Mlawer et al., 1997) longwave scheme; Noah-MP land surface model (Niu et al., 2011); and revised MM5 surface layer scheme (Jimenez et al., 2012).

In the present study, different statistical indicators have been used for the model evaluation with respect to observations/reanalysis datasets. Statistical parameters such as mean absolute error (MAE), root mean square error (RMSE), mean bias (MB), index of agreement (IOA), and correlation coefficient (CC) are defined as:

1. Mean absolute error:

$$MAE = \frac{\sum_{i=1}^{n} |p_i - o_i|}{n}$$

2. Root mean square error:

$$RMSE = \sqrt{\frac{\sum_{i=1}^{n} (p_i - o_i)^2}{n}}$$

3. Mean bias

$$MB = \overline{(p_i - o_i)}$$

4. Index of agreement

$$IOA = 1 - \frac{\sum_{i=1}^{n} (o_i - p_i)^2}{\sum_{i=1}^{n} (|p_i - \bar{o}| + |o_i - \bar{o}|)^2}$$

5. Correlation coefficient

$$CC = \frac{\sum_{i=1}^{n} (p_i - \bar{p})(o_i - \bar{o})}{\sqrt{\sum_{i=1}^{n} (p_i - \bar{p})^2} \sqrt{\sum_{i=1}^{n} (o_i - \bar{o})^2}}$$

in which $p_i$ and $o_i$ represent the predicted and observed time series, respectively, while and $\bar{p}$ and $\bar{o}$ are the predicted and observed mean for a considered variable, respectively.

6. Taylor diagram: It exhibits how well patterns match each other in terms of their correlation, ratio of their variances, and root mean square differences (Taylor, 2001).

7. Q-Q plot: It is a graphical technique used to compare the overall distribution of predicted and observed values for a variable (Venkatram, 1999)

The error or deviation between observed and simulated values is measured by MAE, RMSE, and MB. On the other hand, IOA is used to assess the trend relationship, or how closely the magnitudes and signs of the observed values are related

to the projected values (Schlunzen and Sokhi 2008). In order to evaluate the spatial patterns with ERA5-Land reanalysis
dataset, statistical metrics such as mean bias (%), RMSE, and pattern correlation (PCC) have been used.
**Code and data availability:** Weather Research and Forecasting Model version 4.2.2 (WRFv4.2.2) is an open source model
and can be downloaded from https://www2.mmm.ucar.edu/wrf/users/download/get_source.html. The model output at the
location of the flux tower at Ranchi (23.412N, 85.440E), India is openly available at https://doi.org/10.5281/zenodo.10435513.
The raw observational data derived from the flux tower at Ranchi (23.412N, 85.440E; India) utilized in the present study can
be obtained from the Indian National Centre for Ocean Information Service upon request
(http://www.incois.gov.in/portal/datainfo/ctczdata.jsp). Hourly ERA5-Land reanalysis data utilized in this study can be found
in its official website https://cds.climate.copernicus.eu/cdsapp#!/dataset/reanalysis-era5-land?tab=form.
**Author contribution:** All authors contributed to the design of the study, analysis, and writing of the manuscript. PN carried
out the computations as well as the analysis of the model output.
**Competing interests:** The authors have declared that they have no conflict of interest.
**Acknowledgements:**
We would like to thank Dr. Manoj Kumar for providing observational data at Ranchi. The authors acknowledge the use of
NCAR-NCL and ERA5-Land reanalysis dataset for this study. The use of supercomputing facility (HPC) provided by IIT
Delhi is gratefully acknowledged. This work is partially supported by INSA, DST, DST-INSPIRE, and YES Foundation. We
wish to thank the reviewers for their helpful comments and suggestions, which have significantly enhanced the quality of this
paper.

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

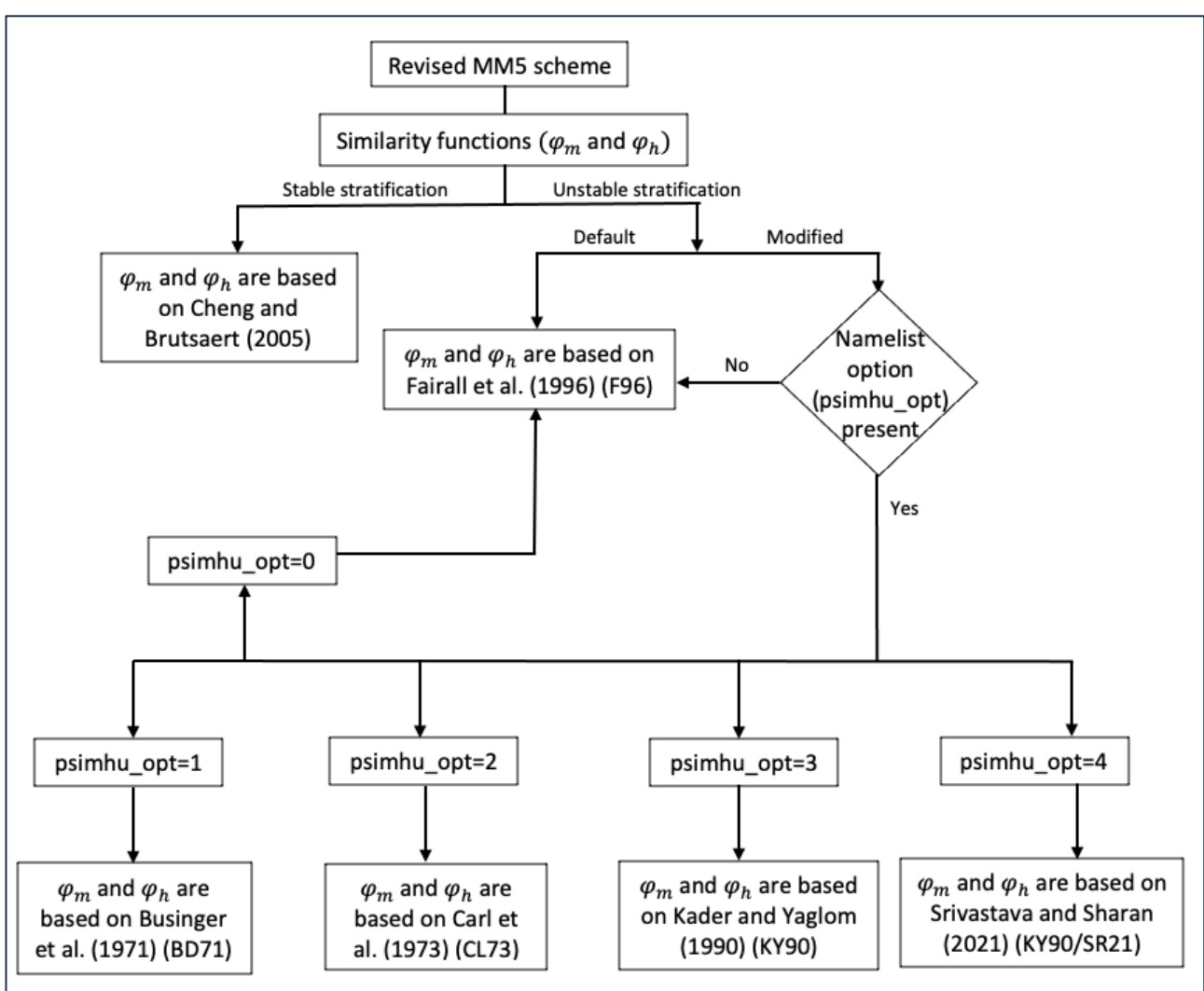

Figure 1: Flowchart to provide a brief description of different options for similarity functions in the modified surface layer scheme that can be controlled by namelist variable psimhu_opt.

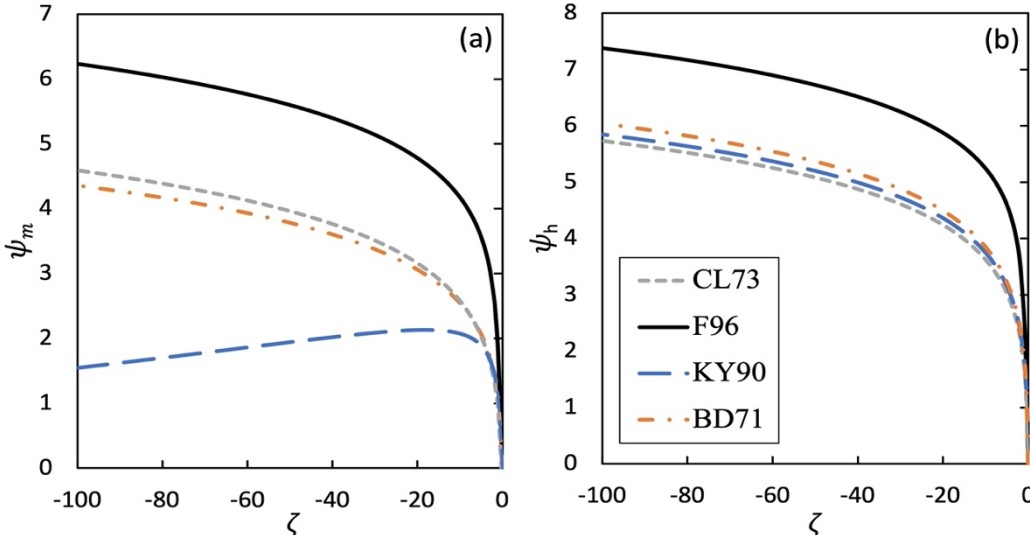

Figure 2: Integrated similarity functions $\psi_{m,h}(\zeta)$ for momentum and heat for default (F96; black line) and newly installed (BD71, CL73, and KY90; orange, grey and blue lines, respectively) functions for unstable atmospheric surface layer.

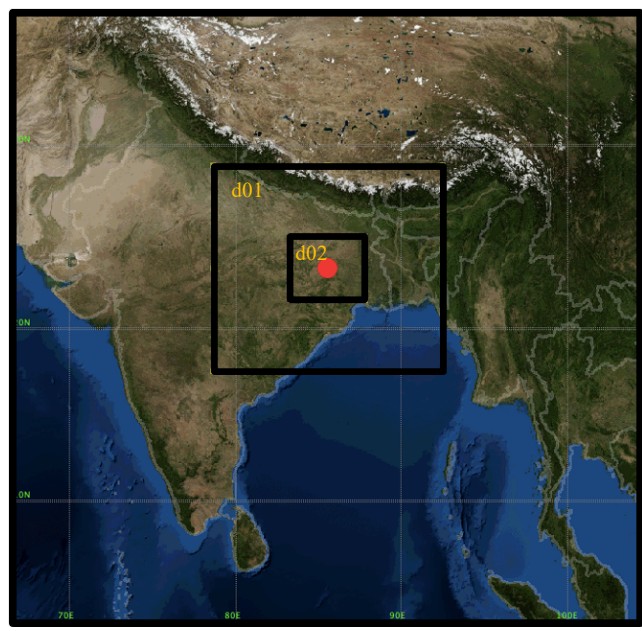

Figure 3: Spatial distribution of domain used for the simulations using the WRF model. The spatial resolution for domains d01 and d02 is $6 \times 6$ km and $2 \times 2$ km, respectively. The domain d02 covers $446 \times 392$ km$^2$ area around the centre point.

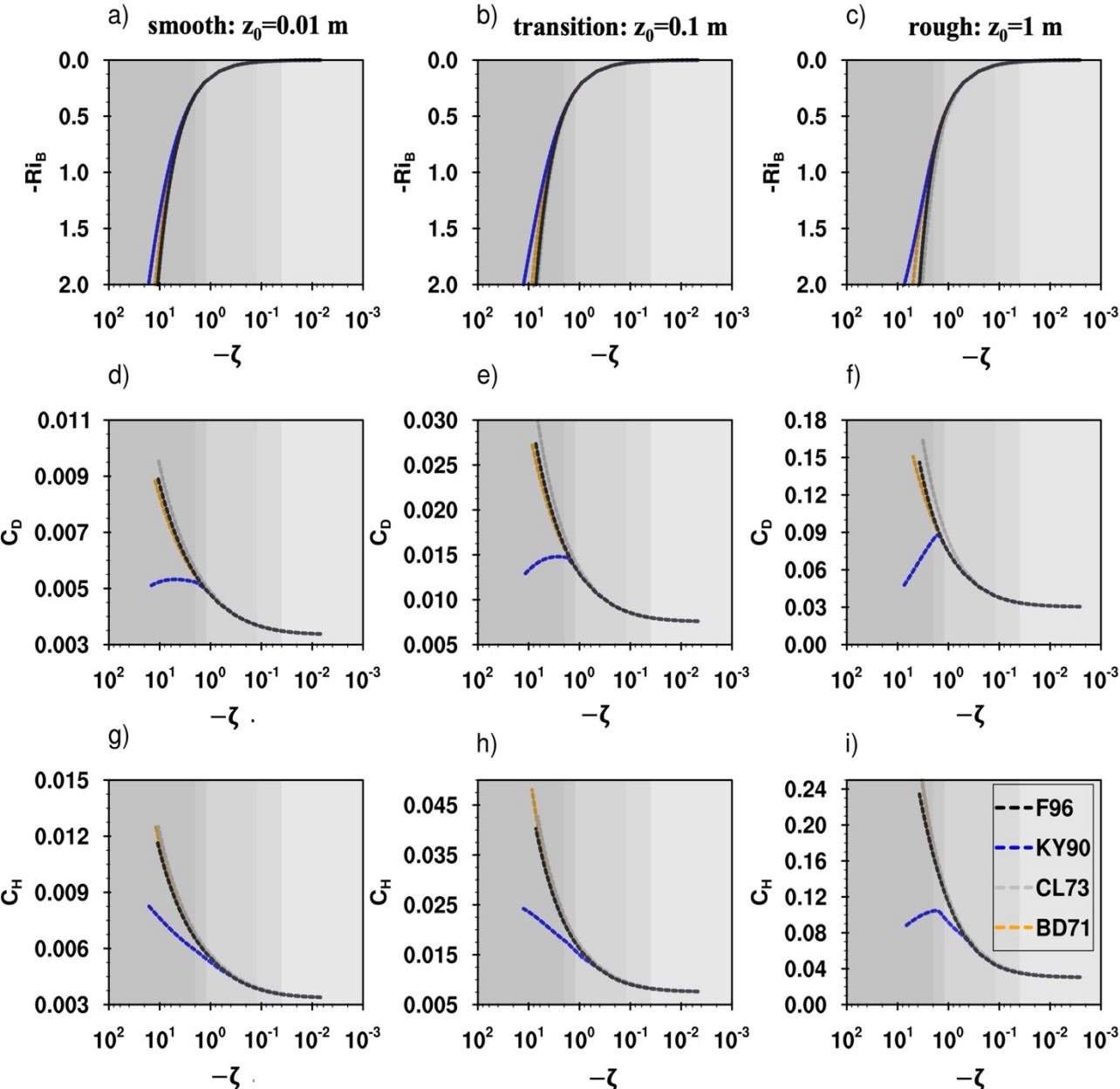

**Figure 4: Variation of ζ with Ri_B (upper panel), C_D (middle panel) and C_H (lower panel) with ζ calculated from bulk flux algorithm (offline simulation) for different functional forms of ψ_m and ψ_h corresponding to BD71, CL73, KY90, and F96 forms for smooth ($z_0 = 0.01$ m; 1st column), transition ($z_0 = 0.1$ m; 2nd column), and rough ($z_0 = 1.0$ m; 3rd column) surfaces. The background colour corresponds to different sublayers in convective conditions (Kader and Yaglom 1990), from the dynamic sublayer ($0 \geq \zeta > -0.04$; light grey) to the free convective sublayer ($\zeta < -2$; dark grey).**




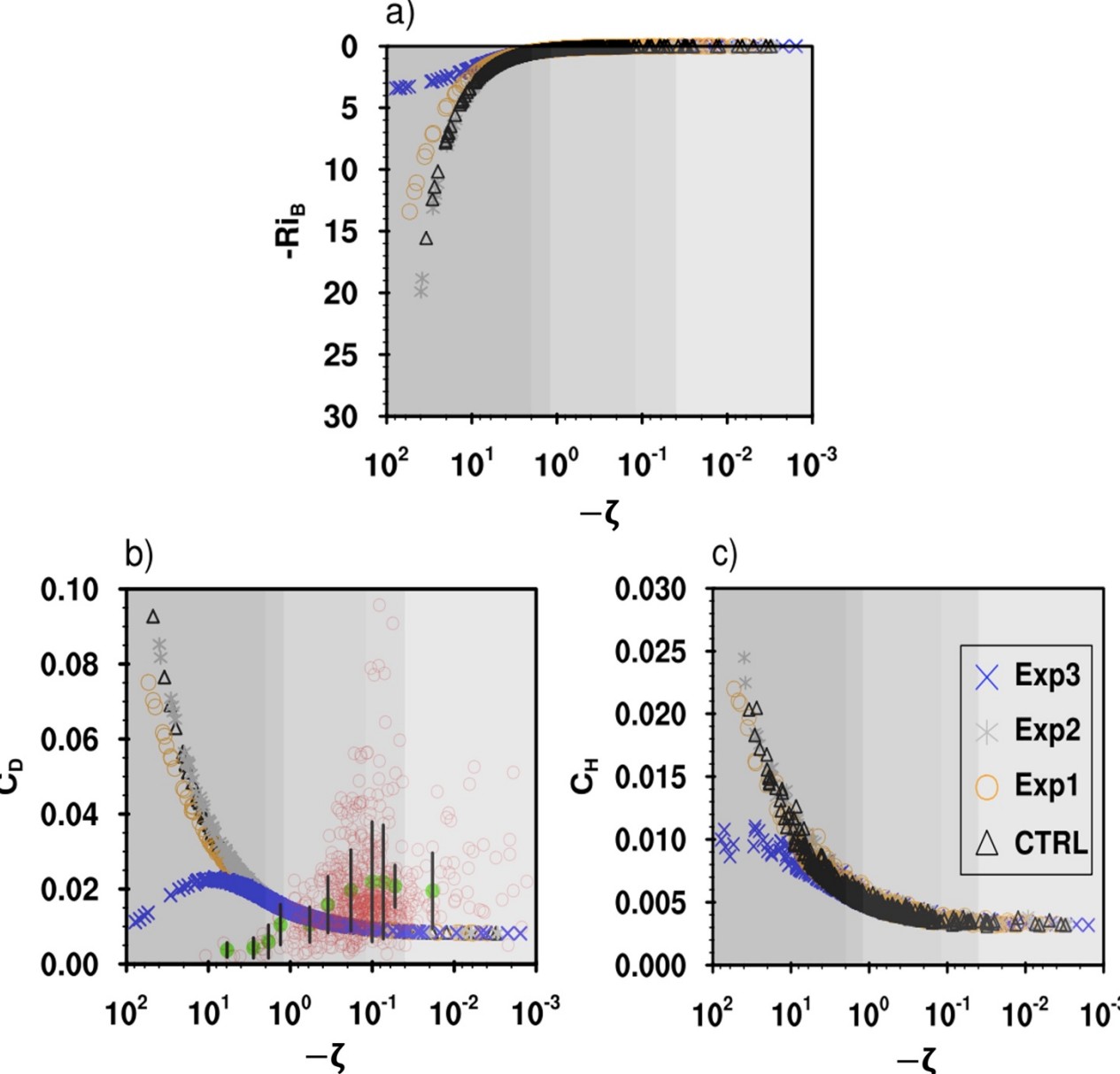


**Figure 5: Variation of model simulated (a) ζ with Ri$_B$, (b) C$_D$ and (c) C$_H$ with ζ from different experiments using different $\psi_m$ and**
**$\psi_h$ corresponding to F96 (CTRL), BD71 (Exp1), CL73 (Exp2), and KY90 (Exp3) under convective conditions. The red circles in (b)**
**denote the observed C$_D$ with ζ at the location of flux tower. The mean values of observed C$_D$ in each sublayer are shown with green**
**solid circles along with standard deviations in the form of error bars. Depending upon the data availability, two or three bins of**
**equal width are chosen in each sublayer. The background colour corresponds to different sublayers in convective conditions (Kader**
**and Yaglom 1990), from the dynamic sublayer ($0 \geq \zeta > -0.04$; light grey) to the free convective sublayer ($\zeta < -2$; dark grey).**

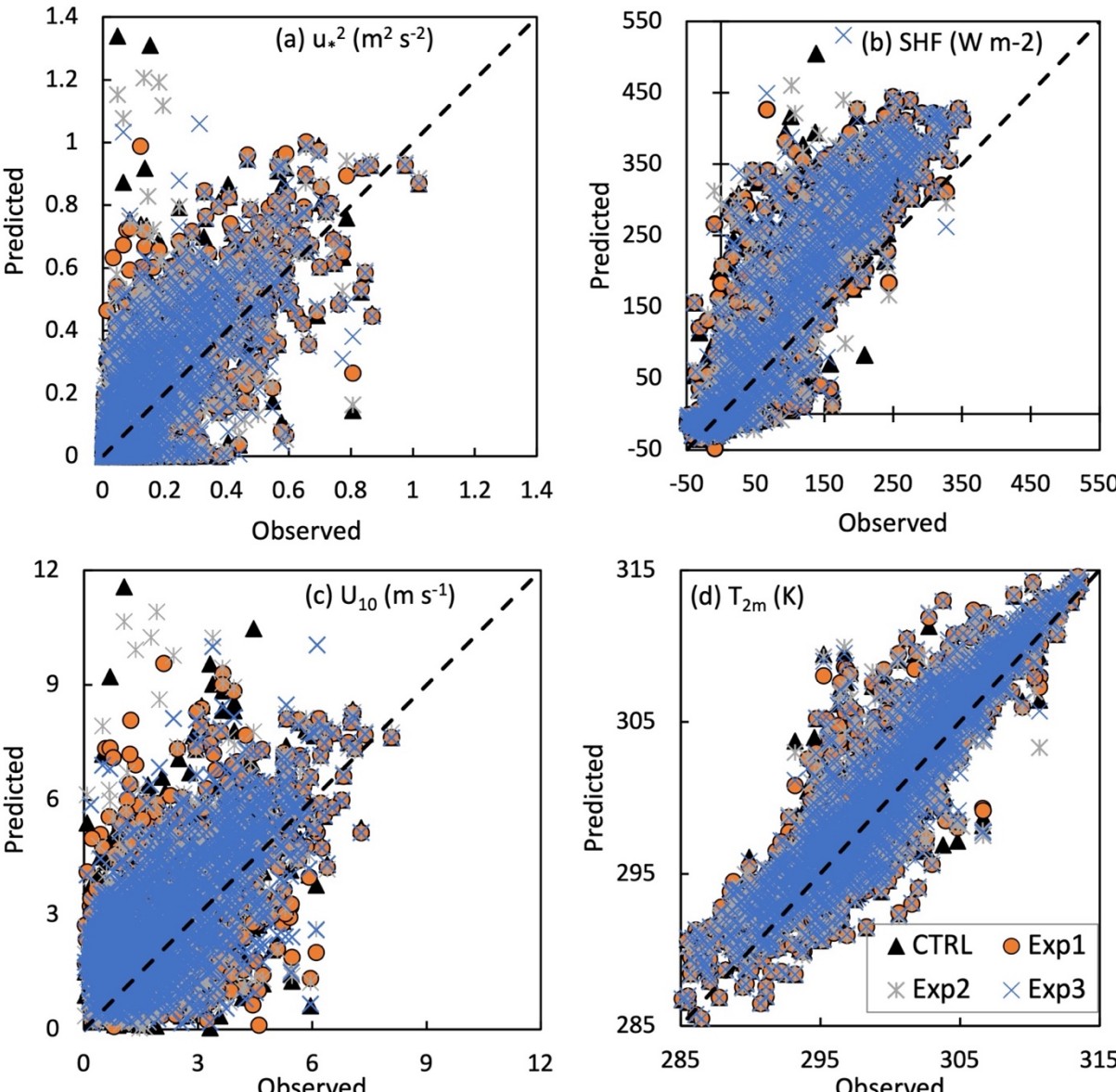

Figure 6: Scatter plot of model simulated (a) $u_*^2$ ($m^2$ $s^{-2}$) (representative of momentum flux), (b) SHF (W $m^{-2}$) (sensible heat flux), (c) $U_{10}$ (m $s^{-1}$) (wind speed at 10 m height), and (d) $T_{2m}$ (K) (temperature at 2 m height) vs observed values at the location of flux tower at Ranchi (23.412oN, 85.440oE), India (centre point of the domain) during pre-monsoon season (MAM).

759

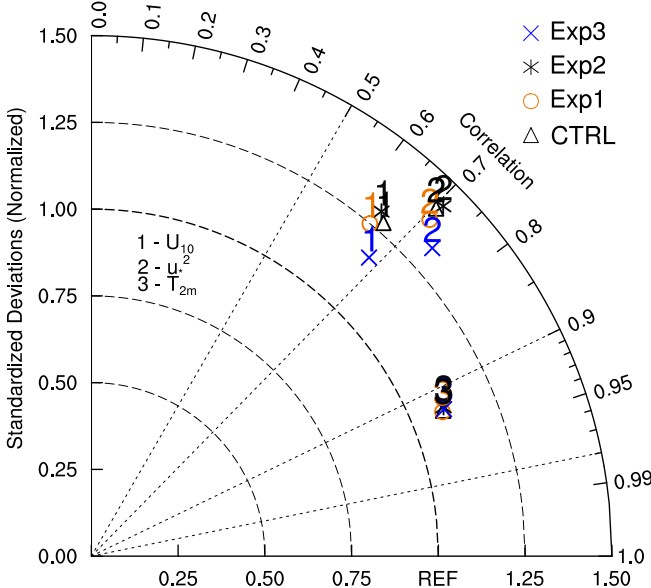

**Figure 7:** Taylor diagram showing the correlation coefficient, normalized standard deviations for $U_{10}$, $u_*^2$, and $T_{2m}$ from different experiments together with CTRL simulation with respect to observations derived from flux tower installed at Ranchi (23.412ºN, 85.440ºE), India.

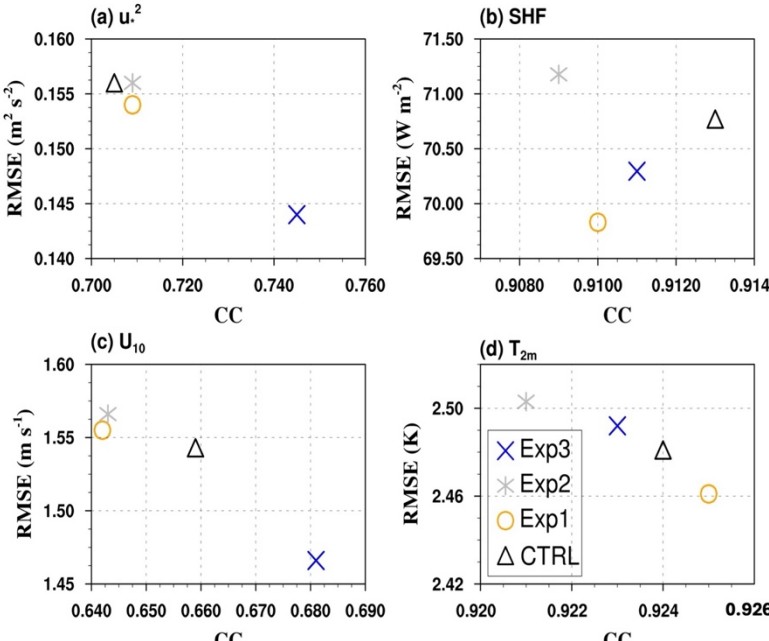

**Figure 8:** Scatter plot between correlation coefficient (CC) and root mean square error (RMSE) for (a) $u_*^2$, (b) SHF, (c) $U_{10}$, and (d) $T_{2m}$ simulated by various experiments (Exp1-3) together with CTRL simulation for pre-monsoon season (MAM; 2009) at the location of the flux tower (23.412ºN, 85.440ºE).

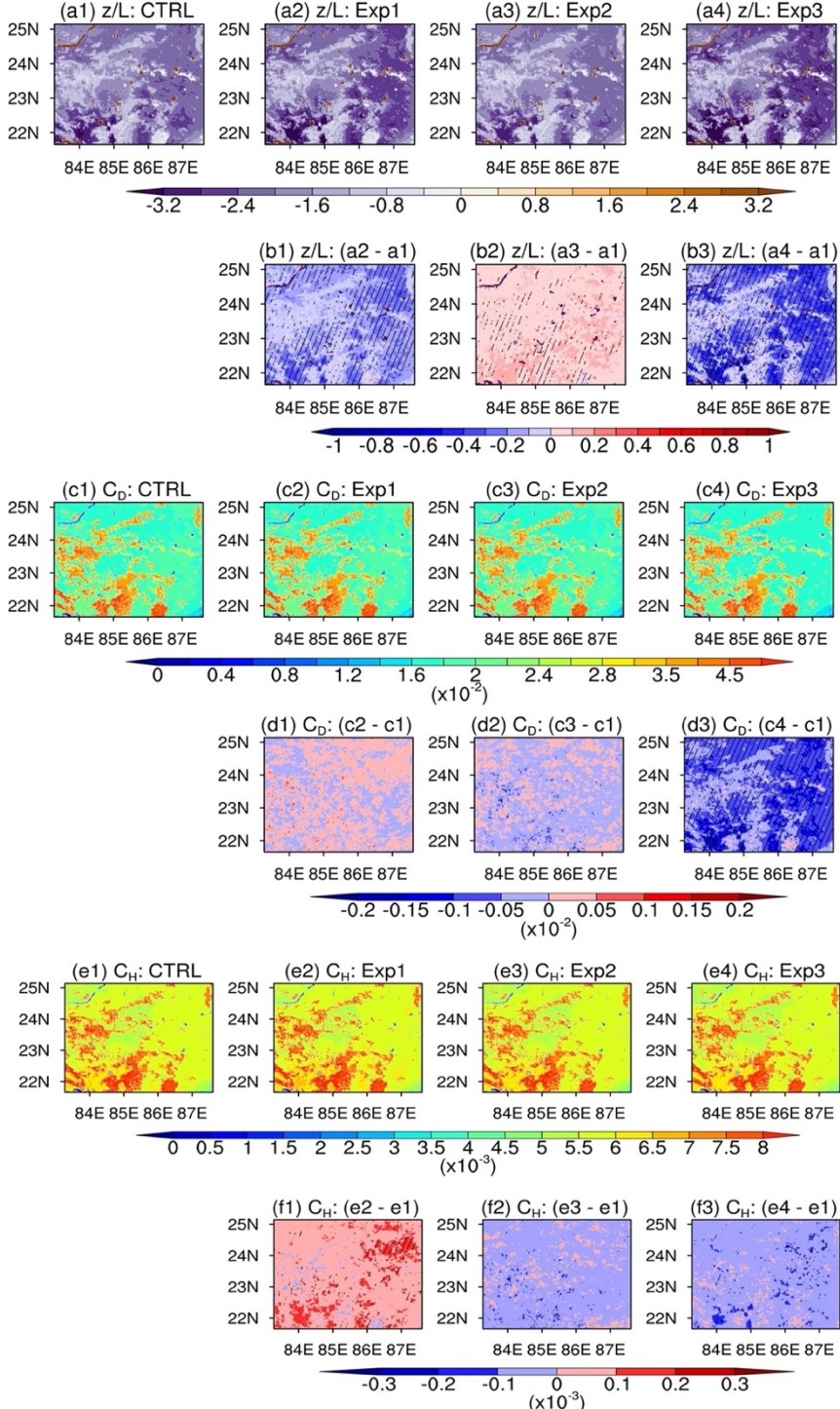

Figure 9: Mean spatial distribution of model simulated ζ (1st row), $C_D$ (3rd row) and $C_H$ (5th row) from different experiments and their differences with respect to CTRL simulation averaged during daytime for whole simulation period. Hatched regions show significant differences at 95% confidence level in experiments with respect to CTRL simulation.

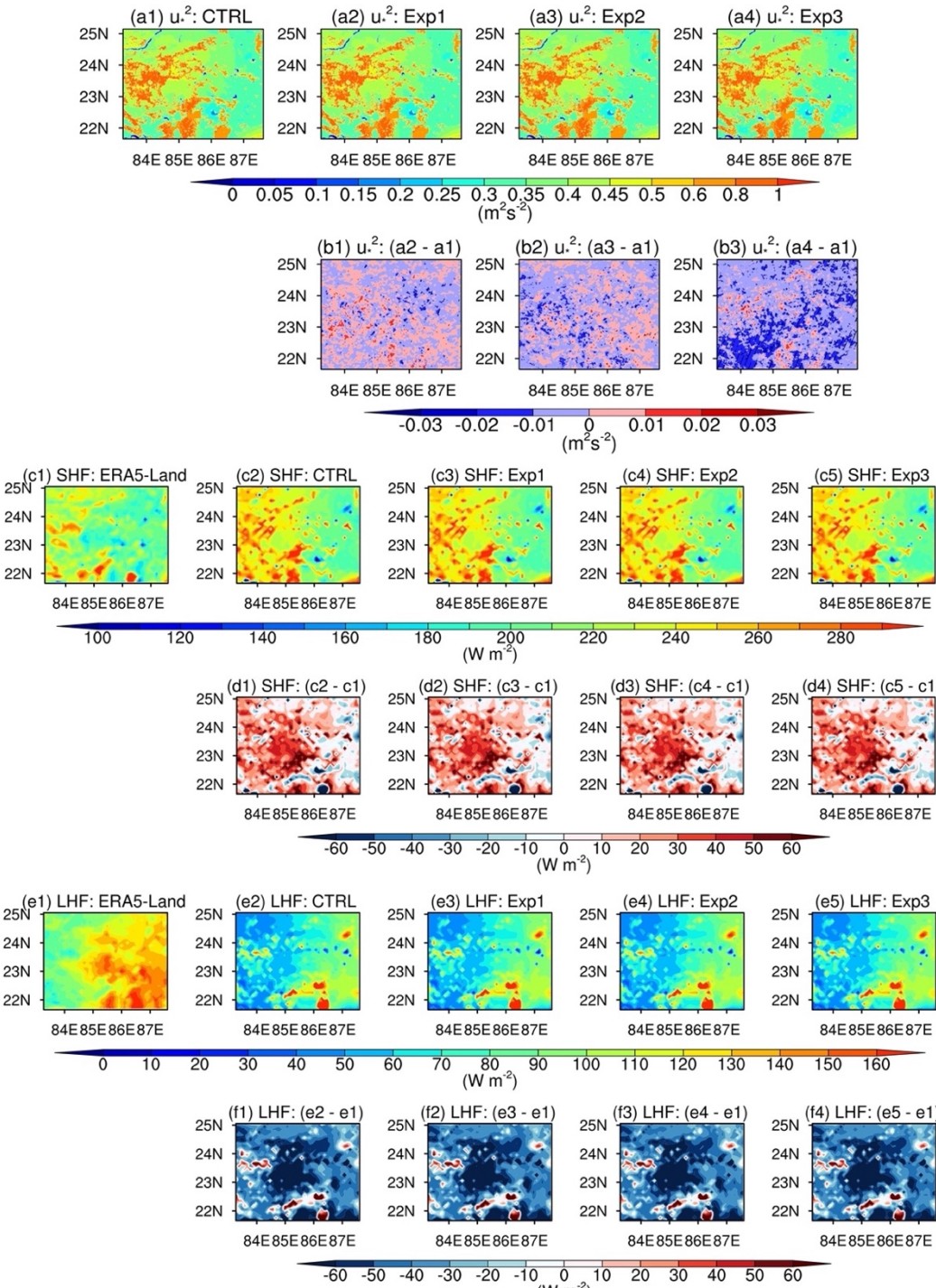

Figure 10: Mean spatial distribution of simulated $u_*^2$ (1st row) from different experiments and their differences (2nd row) with respect to CTRL simulation. SHF and LHF from ERA5-Land reanalysis and simulated using various experiments and their differences with respect to ERA5-Land data averaged during daytime for the whole simulation period are shown. Hatched regions show significant differences at 95% confidence level in experiments with respect to CTRL simulation.

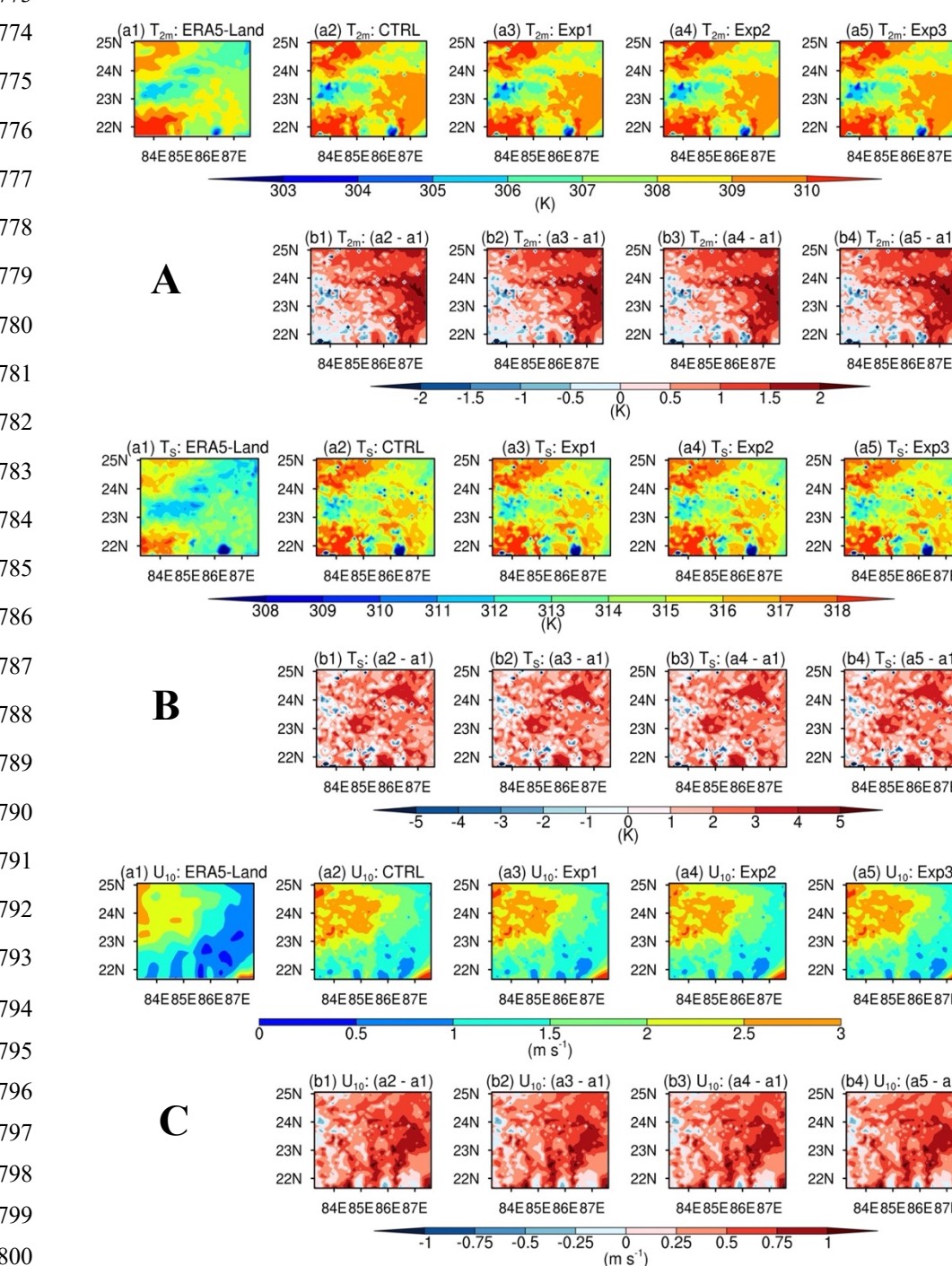

Figure 11: In upper panel (A), mean spatial distribution of $T_{2m}$ from ERA5-Land reanalysis (a1) and simulated using different experiments (a2-a5) and their differences with respect to ERA5-Land reanalysis (b1-b4) averaged during daytime for the whole simulation period. Middle (Lower) panel is same as the upper panel but for $T_S$ ($U_{10}$).

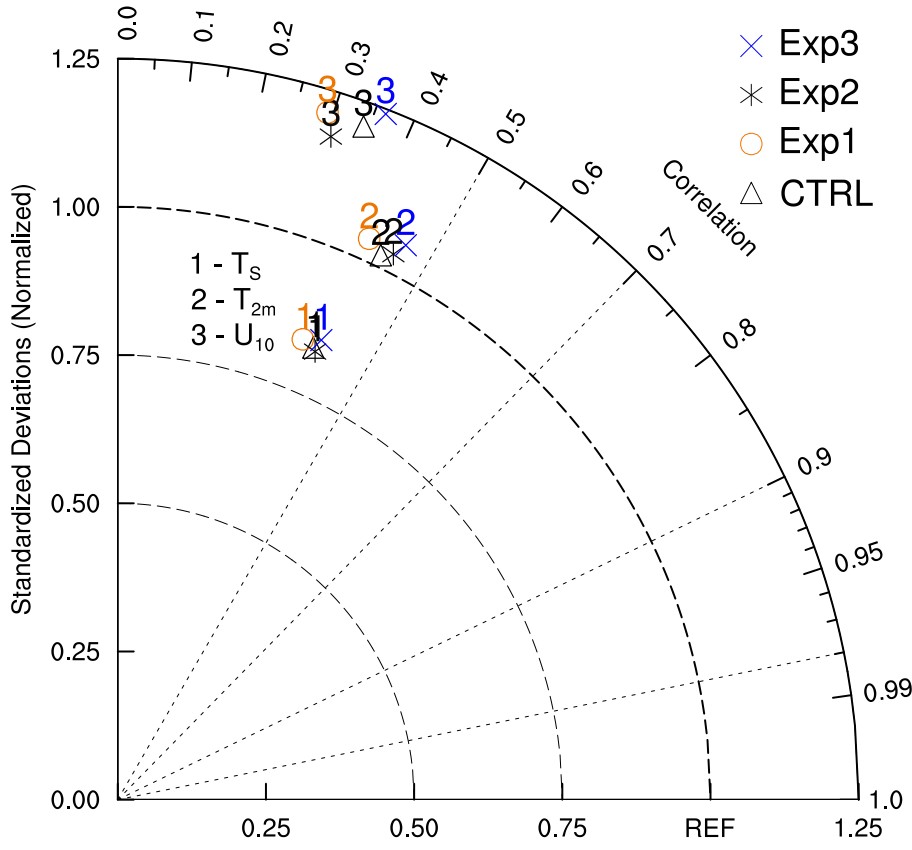

Figure 12: Taylor diagram showing the correlation coefficient, normalized standard deviations for $T_S$ (K), $T_{2m}$ (K), and $U_{10}$ (m s$^{-1}$) from different experiments together with CTRL simulation with respect to ERA5-Land reanalysis dataset averaged during strong convective conditions (hours during daytime in which $\zeta$ is smaller than $-10$) for whole simulation period.

**Table 1. Description of various simulations conducted in this study.**

| Experiments | Description |
| --- | --- |
| CTRL | Simulation using default surface layer scheme with F96 functions |
| Exp1 | Simulation using surface layer scheme with BD71 functions |
| Exp2 | Simulation using surface layer scheme with CL73 functions |
| Exp3 | Simulation using surface layer scheme with newly installed KY90 functions |


**Table 2: Comparison statistics for $u_*^2$ (m² s⁻²), SHF (W m⁻²), U₁₀ (m s⁻¹), and T₂ₘ (K) simulated using different experiments together**
**with CTRL simulation with respect to observations derived from flux tower at Ranchi (India) for MAM season. The mean absolute**
**error (MAE), root mean square error (RMSE), mean bias (MB), index of agreement (IOA), and correlation coefficient (CC) are**
**shown.**

| MAM | | $u_*^2$ (m² s⁻²) | SHF (W m⁻²) | U₁₀ (m s⁻¹) | T₂ₘ (K) |
|---|---|---|---|---|---|
| **CTRL** | **MAE** | 0.09 | 43.46 | 1.20 | 1.82 |
| | **RMSE** | 0.16 | 70.77 | 1.54 | 2.48 |
| | **MB** | 0.03 | 34.88 | 0.83 | 0.93 |
| | **IOA** | 0.82 | 0.89 | 0.73 | 0.95 |
| | **CC** | 0.71 | 0.91 | 0.66 | 0.92 |
| **Exp1** | **MAE** | 0.09 | **42.72** | 1.20 | **1.81** |
| | **RMSE** | 0.15 | **69.83** | 1.56 | **2.46** |
| | **MB** | 0.03 | **33.06** | 0.81 | 0.90 |
| | **IOA** | 0.82 | 0.89 | 0.72 | **0.96** |
| | **CC** | 0.71 | 0.91 | 0.64 | **0.93** |
| **Exp2** | **MAE** | 0.09 | 43.55 | 1.20 | 1.84 |
| | **RMSE** | 0.16 | 71.18 | 1.57 | 2.50 |
| | **MB** | 0.03 | 34.49 | 0.81 | **0.87** |
| | **IOA** | 0.82 | 0.89 | 0.72 | 0.95 |
| | **CC** | 0.71 | 0.91 | 0.64 | 0.92 |
| **Exp3** | **MAE** | **0.08** | 42.96 | **1.16** | 1.83 |
| | **RMSE** | **0.14** | 70.30 | **1.47** | 2.49 |
| | **MB** | **0.03** | 33.47 | **0.78** | 0.91 |
| | **IOA** | **0.84** | 0.89 | **0.75** | 0.95 |
| | **CC** | **0.74** | 0.91 | **0.68** | 0.92 |











Table 3: Comparison statistics for $T_{2m}$ (K), $T_S$ (K), and $U_{10}$ (m s$^{-1}$) simulated using different experiments together with CTRL simulation with respect to ERA5-Land reanalysis data averaged during strong unstable stratification (hours during daytime in which $\zeta$ is smaller than $-10$) for whole simulation period. The percent mean bias (Bias %), pattern correlation coefficient (PCC), and root mean square error (RMSE) are shown.

| MAM | $T_S$ (K) | | | $T_{2m}$ (K) | | | $U_{10}$ (m s$^{-1}$) | | |
|---|---|---|---|---|---|---|---|---|---|
| | Bias (%) | RMSE | PCC | Bias (%) | RMSE | PCC | Bias (%) | RMSE | PCC |
| CTRL | 1.26 | 4.01 | 0.40 | 0.64 | 2.13 | 0.43 | -4.96 | 0.44 | 0.34 |
| Exp1 | 1.26 | 4.03 | 0.37 | 0.64 | 2.16 | 0.40 | -4.43 | 0.45 | 0.29 |
| Exp2 | 1.25 | 3.99 | 0.40 | 0.63 | 2.10 | 0.45 | -5.39 | 0.44 | 0.31 |
| Exp3 | **1.24** | **3.97** | **0.41** | **0.62** | **2.10** | **0.46** | **-0.28** | 0.47 | **0.36** |