# Peer review of "An Updated Parameterization of the Unstable Atmospheric Surface Layer in the WRF Modeling System"

_Geoscientific Model Development, 2024_

## Author Comment (AC1)

**Response to the comments from Reviewer#1**

We thank the reviewer for his critical evaluation of the manuscript, as well as his valuable comments and suggestions. A point-wise reply to the comments of the reviewer is given below:

**General Comment**

*Numerical weather prediction and climate models use surface flux parameterizations depending on Monin Obukhov similarity functions. Various versions of the functions exist in the literature and this paper investigates their effect on model results by using them in the WRF model. The latter is applied to a limited area in the tropics, for which surface flux data exist (the Ranchi data). The focus is on convective conditions. By comparing, e.g. model output with the observations the authors conclude finally that a certain set of functions proposed by Kader and Yaglom (1990) showing non-monotonic behaviour in unstable conditions is superior to other functions. Several sets of functions have been newly implemented by the authors in the model.*

*The topic is important for model applications but also for theoreticians. I find the paper interesting but the text needs better adjustment to the results shown in the figures. I expect that after such modification the study can be published finally. Before that, some paragraphs (also figures) should be improved and some points need clarification. Perhaps, several unclear issues arose due to language problems, so that I also recommend English editing before publication.*

**Reply:** We thank the reviewer for carefully going through the manuscript and giving useful comments and suggestions, as well as encouraging remarks.

**Major Revisions**

*Comment 1: My most important point refers to the differences between the results obtained with different sets of stability functions. To my mind the authors are overinterpreting the differences between the results seen in Figure 4. In my opinion the main finding is here that results obtained by CTRL and experiments 1, 2 and 3 are very similar, when the absolute value of zeta is smaller than about 1.5. Results from CTRL, exp1 and exp2 are even similar in the whole range of zeta. I think, relative to the scatter of observations, only results obtained with exp3 differ really strongly from all other results, but also only when the absolute value of zeta is larger than about 1.5. The discussion of results and conclusions should be reformulated in this direction to reflect the figures 4. The small differences explain also why most results seen in Figures 9,10,11 for different stability functions are so similar. Also here the present text suggests something else.*

**Reply:** We sincerely agree with the reviewer's concern. It is true that experiments 1, 2, and 3 (Exp1-3) and the CTRL simulation using the default version of the surface layer scheme in the WRF model seem to be identical when $\zeta$ lies between 0 and $-1.5$ (approx.), as stated in section 4.1. Moreover, we also agree with the fact that at higher instabilities, only Exp3 shows substantial differences, while Exp1, 2, and CTRL simulations are found to be nearly identical.

As per the reviewer's suggestions, we modified the text in the revised version of the manuscript for better clarity and presentation of the results for general readers. The modified text is presented here for reviewers' reference:

**Lines 233-237…**It is observed that the simulated values of $\zeta$ at smaller values of $Ri_B$ (i.e., in DNS to DCS) from different forms of similarity functions are found to be identical to the F96 forms of functions (Figure 4a1-3). Moreover, results from the BD71, CL73, and F96 functions are even similar at higher instabilities (i.e., the whole range of $\zeta$ values), while substantial differences have been observed in the simulated values of $\zeta$ for a given $Ri_B$ from Exp 3 (Figure 4a1-3).

Moreover, we also agree with the fact that the results obtained from Exp3 differ really strongly from all other results, but only when the absolute value of zeta is larger than about 1.5. The related text is modified accordingly in the revised version of the manuscript and stated here for reviewers' reference:

**Lines 277-282…**The values of simulated variables are found to be almost identical in DNS to DCS sublayers for all the experiments. Moreover, in FCS, the results obtained from Exp1, 2 and CTRL simulation are found to be nearly similar; however, relatively strong differences have been observed in results from Exp 3 (Figures 5a, b, and c). Simulated $\zeta$ for a given $Ri_B$ in Exp2 and CTRL simulation are similar and found to be relatively smaller in magnitude than Exp1 and Exp3 in FCS. However, the absolute values of $\zeta$ in Exp3 (KY90 functions) are relatively higher in FCS than in all other experiments.

Moreover, the comparison of model simulated variables, namely (a) $u_*^2$ ($m^2\ s^{-2}$) (representative of momentum flux), (b) SHF (W $m^{-2}$) (sensible heat flux), (c) $U_{10}$ (m $s^{-1}$) (10-m wind speed), and (d) $T_{2m}$ (K) (2-m temperature), with the observed data obtained from the flux tower at Ranchi (23.412N, 85.440E; India) presented in section 4.2.1 clearly highlights that only Exp3 shows strong differences in the simulated values than all other experiments as well as CTRL simulation.

Figures 9, 10, and 11, which are used to evaluate the mean spatial distribution of simulated variables against ERA5-Land reanalysis data during daytime. While we sincerely agree that the differences between the results presented in Figures 9, 10, and 11 are small, we want to emphasize that some variables (hatched regions of Figures 9, 10, and 11) show significant differences at the 95% confidence interval. The corresponding text has been modified accordingly in the revised version of the manuscript. The modified text is presented here for reviewers' reference:

**Lines 354-357…**It is observed that the absolute value of $\zeta$ simulated in Exp3 (KY90 functions) is relatively smaller than CTRL simulation (Figure 9b3) across the whole domain, which is consistent with Figure 5a and offline simulations presented in Figure 4(a1-3). This could be because the magnitude of the KY90 functions ($\varphi_m$ and $\varphi_h$) is relatively smaller than the functions employed in the default scheme (CTRL simulation).

**Lines 359-366…**Model simulated $C_D$ is found to be relatively smaller in Exp3 than CTRL simulation (Figure 9d3), while Exp1 and Exp2 provide comparable values of $C_D$ to CTRL simulation (Figure 9d1-2). In the case of $C_H$, the simulated values from different experiments are observed to be comparable to the CTRL simulation over the whole study domain (Figure 9f1-3). Note that simulated $C_H$ is found to be comparable in all the experiments, while slight differences have been observed in $C_D$ in Exp 3 compared to all other experiments, which may be related to the fact that only $\varphi_m$ functions are involved in the computation of $C_D$ (Eqn. 1), and the differences between $\varphi_m$ corresponding to Exp3 are relatively more than $\varphi_h$, so are the differences in $C_D$. The hatched regions in Figure 9 show that the differences between simulated variables from different experiments with respect to CTRL simulation are statistically significant at the 95% confidence level.

**Lines 367-372…**The slight differences in $C_D$ in Exp3 were further reflected in the simulated $u_*^2$ $m^2$ $s^{-2}$ (a measure of momentum flux) (Figure 10b3). A slight reduction has been observed in simulated $u_*^2$ in Exp3 compared to the CTRL simulation over some parts of the domain (Figure 10b3), while in Exp1 and Exp2 values are comparable with the CTRL simulation (Figure 10b1-2). In case of SHF and LHF, the mean spatial distribution from all the experiments is found to be consistent with the ERA5-Land reanalysis data, and the magnitude of differences between model simulation and ERA5-Land data is comparable for all the experiments (Table S1; supplementary material).

*Comment 2: I think that the differences between the offline simulation and what is called here 'real-case' simulations using four sets of stability correction functions (default scheme, BD71, CL73, KY90) in WRF should be made clearer. E.g. I recommend to avoid the expression 'real' in this connection and to replace the heading 'Real Case Simulations' by something like 'Results of WRF using different sets of integrated stability correction functions'.*

**Reply:** As suggested by the reviewer, we have replaced the heading 'Real-Case Simulations' by 'Results of WRF using different sets of integrated similarity functions' in the revised version of the manuscript as suggested by the reviewer.

*Comment 3: The offline-simulations would become clearer, if they were not called 'experiment'. What we see in the figure, are the functional dependences of several*

*parameters from stability and surface roughness. The wording 'experiment' is more appropriate for the different model applications.*

**Reply:** We agree with the reviewer's suggestion about the offline simulations. The text related to this has been modified in the revised version of the manuscript and presented here for reviewers' reference:

[revised manuscript text omitted]

Moreover, we wish to highlight that the figure corresponding to offline simulation has also been modified accordingly and presented here for reviewers' reference:

[Figure]

**Figure 4:** Variation of $\zeta$ with $Ri_B$ (upper panel), $C_D$ (middle panel) and $C_H$ (lower panel) with $\zeta$ calculated from bulk flux algorithm (offline simulation) for different functional forms of $\psi_m$ and $\psi_h$ corresponding to BD71, CL73, KY90, and F96 forms for smooth ($z_0 = 0.01$ m; 1st column), transition ($z_0 = 0.1$ m; 2nd column), and rough ($z_0 = 1$ m; 3rd column) surfaces. The background colour corresponds to different sublayers in convective conditions (Kader and Yaglom 1990) from dynamic sublayer ($0 > \zeta > -0.04$; light grey) to free convective sublayer ($\zeta < -2$; dark grey).

*Comment 4: I wonder also why the observations (Figure 5a) are not shown already in Figure 4. If the different surface roughnesses are the reason, this must be explained. Also, the reader should know why in Figure 5 Ch shows variability for a given zeta, but no scatter is seen in Cd. I guess, the reason is the parametrization of the ratio of momentum roughness and scalar roughness, but this must be said. Which are the values for the observations and which parameterization is used for this ratio in WRF?*

**Reply:** We have not shown observational data for $C_D$ in Figure 4 since Figure 4 is used to describe the dependence of estimated $\zeta$, $C_D$, and $C_H$ on different functional forms of similarity functions in a theoretical framework, and we have estimated these variables for three different values of momentum roughness length ($z_0$), which are representative of smooth ($z_0 = 0.01$ m), transition ($z_0 = 0.1$ m), and rough ($z_0 = 1.0$ m) surfaces. Since the observational data site has a different roughness length for momentum, we have not included the observed $C_D$ in Figure 4.

To the best of our knowledge, the WRF model utilizes constant values for roughness length, and the momentum and scalar roughness lengths are assumed to be similar over the land surface. However, the relatively large scatter in the values of $C_H$ simulated from the WRF model may be linked with the fluctuations in the temperature difference term ($\theta_a - \theta_g$).

As per the reviewer's suggestion, text has been added in the revised version of the manuscript and presented here for the reviewers' reference:

**Lines 260-264…**Note that Figure 4 is used to describe the dependence of estimated $\zeta$, $C_D$, and $C_H$ on different functional forms of similarity functions in a theoretical framework, and we have estimated these variables for three different values of momentum roughness length ($z_0$), which are representative of smooth ($z_0 = 0.01$ m), transition ($z_0 = 0.1$ m), and rough ($z_0 = 1.0$ m) surfaces. Since the observational data site has a different roughness length for momentum, we have not included the observed $C_D$ in Figure 4.

**Lines 310-314…**We wish to point out that a relatively larger scatter has been observed in the values of $C_H$ than $C_D$. To the best of our knowledge, the WRF model utilizes constant values for roughness length, and the momentum and scalar roughness lengths are assumed to be similar over the land surface. However, the relatively large scatter in the values of $C_H$ simulated from the WRF model may be linked with the fluctuations in the temperature difference term ($\theta_a - \theta_g$).

*Comment 5: Considering Figure 5a) it seems that the stability range, for which KY results diverge from other results, does not occur in nature, but at least not in the Ranchi data. This should be stressed. Are there other observations, which show a better agreement with the used functions? This should at least be discussed.*

**Reply:** Indeed, the Ranchi data does not display the stability range where the KY90 functions' results diverge from other similarity functions. Due to the limited availability of observational

datasets, the validity and practical applicability of these functional forms to reduce the uncertainties in surface flux formulations are limited, specifically over Indian land. In this study, we wish to compare the performance of the four different functional forms of similarity functions under convective conditions in the surface layer scheme of the WRF model with respect to the Ranchi dataset. It is observed that the KY90 functional forms are the only ones among all the considered functions that can predict $C_D$ consistent with its observed non-monotonic behaviour over Indian land. At the same time, the study also highlights that the $C_D$ predicted from the original forms of KY90 functions shows large disagreement with the observed data, as the predicted $C_D$ starts decreasing at $\zeta$ lying in FCS, which is different from that observed, i.e., $\zeta$ lying in DCS. In light of this, the study further pointed out the need to tune the original form of KY90 functions.

Note that a study by Srivastava and Sharan (2021) attempted to tune the original forms of KY90 functions by enforcing the matching of the point at which both observed and model predicted $C_D$ attain their maximum value. However, more studies in terms of predicting the observed variation of the non-dimensional vertical gradients of mean wind speed and temperature with $\zeta$ are essential to further tune the original KY90 functions for the Indian region using observed data from various locations and seasons.

We wish to highlight that the KY90 functional forms show relatively better agreement with the Ranchi dataset for $u_*^2$ (a representative of momentum flux) and $U_{10}$ (10-m wind speed) and found comparable for other considered variables when employed in the surface layer scheme of the WRF model.

Further, we would like to point out that currently no observational datasets are available that show better agreement with the KY90 functions over Indian land. However, it is desirable to validate these functional forms over Indian land with respect to such observational datasets if they are available in the future.

As suggested by the reviewer, we have added text to the revised version of the manuscript to discuss this issue and presented it here for the reviewer's reference:

**Lines 292-304...**However, it is found that the $C_D$ predicted from the original forms of class 4 functions (Exp3) shows large disagreement with its observed behaviour, as the predicted $C_D$ starts decreasing at $\zeta$ lying in FCS, which is different from that observed, i.e., $\zeta$ lying in DCS. In light of this, the study further pointed out the need to tune the original form of KY90 functions and the need for more studies to further evaluate the performance of the original forms of KY90 functions in the WRF model using different available observational datasets from different Indian land sites and seasons.

Note that Srivastava and Sharan (2021) tuned the original forms of class 4 functions by enforcing the matching of the point at which both observed and model predicted $C_D$ attain their maximum value. However, more studies in terms of predicting the observed variation of the non-dimensional vertical gradients of mean wind speed and temperature with $\zeta$ are essential to further tune the original KY90 functions for the Indian region using observed data from various locations under different seasons.

Further, we would like to point out that currently no observational datasets are available that show a better agreement with the KY90 functions over Indian land. However, it is desirable to validate these functional forms over Indian land with respect to such observational datasets if they are available in the future.

**Other recommendations:**

*Comment 1: Line 136: all this would be more convincing if data would be added in Figures S1a and b, Also, the reader would like to know if KD had perhaps physical arguments for proposing non-monotonous functions.*

**Reply:** Figures S1a and b show the variation of the newly installed (BD71, CL73, and KY90) and default $\varphi_m$ and $\varphi_h$ with respect to $-\zeta$ under convective conditions. Thus, we have just plotted different $\varphi_m$ and $\varphi_h$ with $\zeta$ in the range $(-100, 0)$. This figure suggests that $\varphi_m$ based on KY90 functional forms exhibits a contrasting behaviour; however, all other functional forms are more or less similar.

Note that the KY90 functional forms are based on the three layer structure of the convective regime proposed by Kader and Yaglom (1990). The dynamic sublayer corresponds to near-neutral conditions in which $\varphi_m = 1$ and $\varphi_h = Pr_t$. Further, in the dynamic convective sublayer, mechanical energy is in the x direction, while buoyancy-induced energy is in the z direction. Thus, in this sublayer, the functional forms for similarity functions, as determined by dimensional analysis, are

$$\varphi_m(\zeta) = A_u(-\zeta)^{-\frac{1}{3}} \qquad (R1)$$

$$\varphi_h(\zeta) = A_T(-\zeta)^{-\frac{1}{3}} \qquad (R2)$$

in which $A_u$ and $A_T$ are constants.

Moreover, in the free-convective sublayer, buoyancy dominates the mechanical production of energy, and the pressure redistribution term feeds the buoyant energy in the vertical direction into the horizontal direction (Kader and Yaglom, 1990). Thus, in this case, the dimensional analysis suggests

$$\varphi_m(\zeta) = B_u(-\zeta)^{\frac{1}{3}} \qquad (R3)$$

$$\varphi_h(\zeta) = B_T(-\zeta)^{-\frac{1}{3}} \qquad (R4)$$

in which $B_u$ and $B_T$ are constants.

Thus, as stated above, in the three layer model, $\varphi_m$ follows a $-1/3$ power law in the dynamic convective sublayer; however, $+1/3$ power in the free convective sublayer is based on the dimensional analysis, which shows a non-monotonic behaviour of $\varphi_m$ with $-\zeta$ under convective conditions. This contradicts the behaviour of $\varphi_m$ predicted by the classical free convection limit.

***Comment 2: Line 167: 'without feedback to the atmosphere'...this formulation might be misleading because this offline simulation is completely independent from WRF. So I would recommend writing something like (starting from line 164): The performance of the default and newly installed similarity functions is investigated in two steps. The first one is independent on the WRF model. Namely, we apply equation A7 to iteratively determine Cd and Ch as a function of zeta by prescribing the bulk Richardson number and surface roughness parameters z_m and z_h...... We call this in the following offline simulation.***

**Reply:** As per the reviewer's suggestions, the text has been modified in the revised version of the manuscript and stated here for the reviewers' reference:

**Lines 190-196…**To analyze the impacts of newly installed similarity functions together with the existing functional forms in the surface layer scheme of WRFv4.2.2, the performance of the default and newly installed similarity functions is investigated in two steps. The first one is independent of the WRF model. Namely, we apply Eqn. (B7) (Appendix B) to iteratively determine $C_D$ and $C_H$ as a function of $\zeta$ by prescribing the bulk Richardson number ($Ri_B$) and surface roughness parameters for momentum ($z_0$) and heat ($z_h$). Note that the values of $z_0$ and $z_h$ are assumed to be same. The value of $\zeta$ is estimated by calculating the root of least magnitude of Eqn. (B7) for a given value of $Ri_B$. Once $\zeta$ is calculated then utilizing it in Eqns. (B8) and (B9), the values of $C_D$ and $C_H$ can be estimated. We call this in the following offline simulation.

***Comment 3: And later in the text, where you start describing the model application. The second step is to apply all parameterizations of the similarity functions in the model WRF whose output is compared then with observations.***

**Reply:** The text has been modified as per the reviewer's suggestion and is presented here for reviewers' reference:

**Lines 200-202…**The second step is to apply all the parameterizations of the similarity functions in the WRF version 4.2.2 model over an Indian land site whose output is compared with the observations during the pre-monsoon (March-April-May; MAM) season of the year 2009.

***Comment 4: Line 203: I do not really see large differences. There is only the function KY90, which produces really large differences to all others. If you think differences between the other functions are also large, this must be better explained. In the present figures I cannot see any relevant difference between results from EXP1, EXP2 and CTRL. There might be a***

*tendency for differences increasing with surface roughness in case of -zeta larger than 1 ? When this is the case, another figure showing this in a zoomed version might be helpful.*

**Reply:** Yes, only the KY90 functional forms show large differences, and all the other functional forms are more or less similar. We have modified the text accordingly in the revised version of the manuscript and presented it here for reviewers' reference:

**Lines 233-237…** It is observed that the simulated values of $\zeta$ at smaller values of $Ri_B$ (i.e., in DNS to DCS) from different forms of similarity functions are found to be identical to the F96 functional forms (Figure 4a1-3). Moreover, results from the BD71, CL73, and F96 functions are even similar at higher instabilities (i.e., the whole range of $\zeta$ values), while substantial differences have been observed in the simulated values of $\zeta$ for a given $Ri_B$ from Exp 3 (Figure 4a1-3).

**Comment 5: Line 210: I do not see that they are really 'substantially' higher (?)**

**Reply:** The related text has been modified in the revised version of the manuscript and is presented here for reviewers' reference:

**Line 244-245…** Notice that the $C_D$ values calculated from the BD71, CL73, and F96 forms of functions are relatively higher than the $C_D$ produced by the KY90 functional forms and continue to rise as instability progresses from DCS to FCS.

**Comment 6: Line 223-225: This last paragraph is the main finding to which I can agree. But this should come earlier, so that this whole subsection could be shortened. But I have another point. Namely, Figure 5 shows that for much stronger instability, differences between all functions become more pronounced. So, why is that not shown already in Section 4.1?**

**Reply:** As stated in the earlier replies, only the KY90 functional forms show large differences in the simulated variables; however, all other functional forms are found to be approximately similar. Figure 5 also suggests that the simulated $\zeta$, $C_D$, and $C_H$ from KY90 functions show large differences for much stronger instabilities; however, all the functions are more or less similar in producing $\zeta$, $C_D$, and $C_H$.

Note that the text has been modified accordingly in the revised version of the manuscript.

**Comment 7: Line 237: it should not be only consistent, but results should be identical if the same roughness parameters are used. Please note that the Rib-zeta curves depend only on height and roughness parameters but not on any other external parameter.**

**Reply:** We agree with the fact that the $Ri_B$-$\zeta$ curves depend only on the height and roughness parameters. We have modified related text in the revised version of the manuscript.

*Comment 8: Line 249: This is not an appropriate formulation. There is a very large disagreement, one should not simply write only that there is no perfect match. Note that the functional forms are completely different.*

**Reply:** The text has been modified and presented here for reviewers' reference:

**Lines 292-294…**However, it is found that the $C_D$ predicted from the original forms of class 4 functions (Exp3) shows large disagreement with its observed behaviour, as the predicted $C_D$ starts decreasing at ζ lying in FCS, which is different from that observed, i.e., ζ lying in DCS.

*Comment 9: Figures must be improved. E.g., in Figure 7, one cannot read the numbers (especially number 3 is unreadable). Please increase also the font size of headings in all figures showing horizontal cross-sections of model results. These headings are almost unreadable without zooming in, for which, however, the resolution is not good enough.*

**Reply:** All the figures have been modified and presented here for reviewers' reference:

[Figure]

**Figure 7:** Taylor diagram showing the correlation coefficient, normalized standard deviations for $U_{10}$, $u_*^2$, and $T_{2m}$ from different experiments together with CTRL simulation with respect to observations derived from flux tower installed at Ranchi (23.412$^O$N, 85.440$^O$E), India.

[Figure]

**Figure 8:** Scatter plot between correlation coefficient (CC) and root mean square error (RMSE) for (a) $u_*^2$, (b) SHF, (c) U10, and (d) T2m simulated by various experiments (Exp1-3) together with CTRL simulation for pre-monsoon season (MAM; 2009) at the location of the flux tower (23.412ᵒN, 85.440ᵒE).

[Figure]

**Figure 9:** Mean spatial distribution of model simulated $\zeta$ (1st row), $C_D$ (3rd row) and $C_H$ (5th row) from different experiments and their differences with respect to CTRL simulation averaged during daytime for whole simulation period. Hatched regions show significant differences at 95% confidence level in experiments with respect to CTRL simulation.

[Figure]

**Figure 10:** Mean spatial distribution of simulated $u_*^2$ (1st row) from different experiments and their differences (2nd row) with respect to CTRL simulation. SHF and LHF from ERA5-Land reanalysis and simulated using various experiments and their differences with respect to ERA5-Land data averaged during daytime for the whole simulation period are shown. Hatched regions show significant differences at 95% confidence level in experiments with respect to CTRL simulation.

[Figure]

**Figure 11:** In upper panel (A), mean spatial distribution of T$_{2m}$ from ERA5-Land reanalysis (a1) and simulated using different experiments (a2-a5) and their differences with respect to ERA5-Land reanalysis (b1-b4) averaged during daytime for the whole simulation period. Middle (Lower) panel is same as the upper panel but for TS (U10).

[Figure]

**Figure 12:** Taylor diagram showing the correlation coefficient, normalized standard deviations for $T_S$ (K), $T_{2m}$ (K), and $U_{10}$ (m s$^{-1}$) from different experiments together with CTRL simulation with respect to ERA5-Land reanalysis dataset averaged during strong convective conditions (hours during daytime in which $\zeta$ is smaller than $-10$) for whole simulation period.

*Minor Revisions*

**Most of these minor revisions refer to language problems, e,g, at many places the English article 'the' is not used correctly or it needs to be added. I give many examples.**

*Comment 1: Line 12: is 'all' really correct, aren't there more such functions?*

**Reply:** It is modified in the revised version of the manuscript. The revised text is presented here for reviewers' reference:

**Line 12…**The surface layer module in WRFv4.2.2 is modified in such a way that it contains the commonly used similarity functions for momentum ($\varphi_m$) and heat ($\varphi_h$) under convective conditions instead of the existing single functional form.

*Comment 2: Line 12: replace 'used phi_m' by 'used similarity functions phi_m' . (The symbols need to be explained at their first occurrence.)*

**Reply:** The needful is done.

*Comment 3:* **Line 28: replace near neutral' by 'near-neutral'. This occurs at many places in the text, I will not repeat it. So please check it.**

**Reply:** The word "near neutral" has been replaced with "near-neutral" throughout the modified text.

*Comment 4: Line 20: do you mean here in the WRF model or in other numerical models as well? The formulation leaves this open. So, what do you mean with 'the ... model? Perhaps a language problem.....*

**Reply:** Yes, it is just for the WRF model, and the text is modified accordingly and is presented here for reviewers' reference:

**Lines 19-20…**The study suggests that the updated surface layer scheme performs well in simulating the surface transfer coefficients and could be potentially utilized for parameterization of surface fluxes in the WRF model.

*Comment 5: Line 52: when you write WRF model, then it should always be 'the model'. Only when you write just WRF (without the word model) then 'the' can be omitted. This arises at many places in the whole text.*

**Reply:** The text has been modified accordingly.

*Comment 6: Line 65: Replace perhaps 'the available' by all available' ?*

**Reply:** The needful is done.

*Comment 7: Line 68: no comma after which*

**Reply:** The needful is done by removing the comma.

*Comment 8: Lines 82-84: The structure is a little puzzling here and I recommend therefore to replace the sentence by something like: Their determination based on MOST using integrated forms of the similarity functions is explained in Appendix A. In the following, the default similarity functions used in WRF are explained and further functions are introduced in Section 2.2.*

**Reply:** The needful is done by modifying the sentence accordingly.

*Comment 9: Line 100: Already here the abbreviation F96 must be introduced, which is used later (?)*

**Reply:** The needful is done by introducing the abbreviation F96.

*Comment 10: Line 102: At this point, the formulation is somehow unclear and I was not yet sure here if these functions are already implemented in WRF or if this implementation is the topic of the paper. It should become clear already here.*

**Reply:** The needful is done, and the modified text is presented here for reviewers' reference:

**Lines 118-121…**In this section, we briefly describe the implementation of different similarity functions under unstable stratification in the surface layer parameterization of WRFv4.2.2. Note that the functional forms suggested by Carl et al. (1973) and the three sub-layer model suggested by Kader and Yaglom (1990) for convective conditions have not been installed and tested in the revised MM5 surface layer in the WRF modeling framework.

*Comment 11: Line 106: replace 'these functions' by 'They'*

**Reply:** The needful is done.

*Comment 12: Line 112: Better write: equations (B3) and (B4) (Appendix B)*

**Reply:** As per another reviewer's suggestion, appendix B is now interchanged with appendix A, and equations (B3) and (B4) are now referred to as equations (A3) and (A4). The text has been modified accordingly in the revised version of the manuscript.

*Comment 13: Line 113: Have not been analyzed by you or by others as well?*

**Reply:** To the best of our knowledge, the functional forms suggested by Carl et al. (1973) have not yet been installed and evaluated in the revised MM5 surface layer scheme of the WRF model.

*Comment 14: Line 114: are given by Equation (6)*

**Reply:** The needful is done.

*Comment 15: Line 127: Mentioning this program parameter is a very specific information for those who are using this model. I suggest describing this more generally or add this very technical description in an appendix.*

**Reply:** The text mentioning this program parameter is already included in the manuscript and is presented here for reviewers' reference:

**Lines 143-148…**Here, we have introduced a new surface layer module where different options for $\varphi_m$ and $\varphi_h$ can be controlled using an appropriate value of the namelist parameter (psimhu_opt). The parameter psimhu_opt is added under the physics section of the namelist file. The variable psimhu_opt can have values 0, 1, 2, and 3 for different options for functions F96 (default), BD71, CL73, and KY90, respectively. Moreover, a brief structure and different choices for psimhu_opt based on newly installed and default functional forms of $\varphi_m$ and $\varphi_h$ in the default and modified revised MM5 scheme are shown in Figure 1.

[Figure]

**Figure 1:** Flowchart to provide a brief description of different options for similarity functions in the modified surface layer scheme that can be controlled by namelist variable psimhu_opt.

*Comment 16: Line 144: replace 'strong' by 'strongly'*

**Reply:** The word "strong" has been replaced by "strongly" in the revised text.

*Comment 17: Line 145: replace 'KY90' by 'the KY90'*

**Reply:** The needful is done.

*Comment 18: Line 148: replace 'other' by 'the other'*

**Reply:** The needful is done.

*Comment 19: Line 148: replace part after functions by: while results of all other functions (BD71, CL73 and KY90) are very similar to each other.*

**Reply:** The text has been modified accordingly.

**Comment 20: Line 164: The wording is not correct: I recommend replacing everywhere in the text (including captions) the formulation 'incorporated functions' by 'newly installed functions' (note that the default function is also an incorporated function in the model).**

**Reply:** The word "incorporated" has been replaced by "newly installed" everywhere in the revised text.

**Comment 21: Line 166: brackets are not correct (see above)**

**Reply:** These are corrected in the modified text.

**Comment 22: Line 185: replace 'of 1st' by 'of the first'**

**Reply:** The needful is done.

**Comment 23: Line 191: replace 'brief' by 'a brief ' and use better' given in' than stated in'**

**Reply:** The needful is done.

**Comment 24: Lines 199-201: Can these sublayers be explained briefly? Not every reader is an expert for this. E.g., what is dynamic convective-free convective?**

**Reply:** The sublayers are explained briefly, and the modified text is presented here for reviewers' reference:

**Lines 230-233…**Note that the sublayers DNS ($-0.04 \leq \zeta \leq 0$) and DNS-DCS transition ($-0.12 \leq \zeta < -0.04$) correspond to weakly to moderately unstable conditions, while sublayers DCS ($-1.20 \leq \zeta < -0.12$), DCS-FCS ($-2.0 \leq \zeta < -1.20$), and FCS ($\zeta < -2.0$) belong to moderately to strongly convective conditions (Srivastava and Sharan, 2015).

**Comment 25: Line 218: This is now in contrast to the description in the preceding paragraph where it is written that differences are large.**

**Reply:** The necessary changes are made.

**Comment 26: Line 238: I guess it should be written almost identical, Identical results can only be achieved, when the same formula is used. This should be corrected at all occurrences.**

**Reply:** The text has been modified accordingly throughout the manuscript.

---

## Author Comment (AC2)

**Response to the comments from Reviewer#2**

We thank the reviewer for his/her critical evaluation of the manuscript. A point-wise reply to the comments of the reviewer is given below:

**General Comment**

**Summary: The paper discusses how the free convection limit needs to be implemented in NWP models, i.e. that fact that in case of vanishing wind speed the friction velocity drops out of the Monin Obukhov scaling. Within the context of the WRF mesoscale model several formulations are discussed and implemented in the surface layer scheme of WRF and tested for a long period of offline and online simulations. It is shown the model is (moderately) sensitive to the selected similarity functions for operational forecasts for a 3 month period.**

**Reply:** We thank the reviewer for carefully going through the manuscript and for his/her valuable comments and suggestions.

**Major Comments**

*Comment 1: Earlier studies, especially the ones done in the GABLS model intercomparison projects have studied the impact of the shape of the stability functions on the modelled profiles and fluxes (though for stable conditions mostly). However, they learnt that applying different stability functions in the surface layer parameterization and in the boundary-layer parameterization may trigger unnatural kinks in the wind speed profiles in models like WRF. This happens in practice quite often in modelling approaches for all kind of reasons. It would be good if the authors can add some discussion about this aspect, and check for (in)consistency of phi-functions in PBL and SL in their updated KY90 formulation. And whether kinks are seen in temperature and wind profiles in the WRF output.*

**Reply:** We sincerely accept the reviewer's concern regarding the unnatural kinks in the wind speed and temperature profiles by using different similarity functions in the surface and boundary layer parameterizations. We wish to highlight that the present study focused on evaluating the impacts of different similarity functions in the revised MM5 surface layer scheme in WRF model version 4.2.2 on the simulation of surface turbulent fluxes and near-surface variables. For the simulations, we have utilized the YSU (Yonsei University) PBL scheme proposed by Hong et al. (2006). This scheme utilizes similarity functions suggested by Dyer (1974) for both unstable and stable conditions in the gradient form; those are different from the ones used in surface layer parameterization.

We have analyzed the behaviour of 10-m wind and 2-m temperature profiles predicted from the WRF model using different forms of similarity functions in the surface layer scheme for the whole simulation period (March-April-May). It is observed that the unnatural kinks

have not been observed in cases of both 10-m wind speed and 2-m temperature (Figures R1 and R2). However, the relatively higher magnitudes of 10-m wind speed simulated from the WRF model have been observed for some hours, which may be linked with the localised weather phenomenon characterized by rapid changes in weather, including strong wind, lightning, and thunderstorms. However, the 2-m temperature values are found to be in line with the observed data. Thus, both 10-m wind speed and 2-m temperature values simulated from the WRF model are justifiable, and no unnatural kinks have been observed. Moreover, further investigation is needed in this direction.

As per the reviewer's suggestion, we have added text in the revised version of the manuscript and presented it here for the reviewers' reference:

**Lines 338-349...**Note that earlier studies, especially the ones done in the GABLS model intercomparison projects, have studied the impacts of the similarity functions on the modelled profiles and fluxes (though for stable conditions mostly). However, they learnt that applying different stability functions in the surface and boundary layer parameterizations may trigger unnatural kinks in the model simulated wind speed and temperature profiles. Here, we have analyzed the profiles of $U_{10}$ and $T_{2m}$ simulated from the WRF model using different similarity functions in the surface layer scheme for the occurrence of unnatural kinks in their values. We observed that the $U_{10}$ predicted from CTRL simulation, along with various experiments corresponding to different similarity functions at specific hours, exceeds its observed maximum value of approx. 8 m s$^{-1}$ (Figure S3). Some localized weather phenomena, such as strong wind, lightning, and thunderstorms, may link these relatively higher magnitudes to rapid changes in weather, making them justifiable. However, the simulated $T_{2m}$ from different similarity functions is found to be in line with the observed values across the whole simulation period (Figure S4). This suggests that the values of $U_{10}$ and $T_{2m}$ predicted from the WRF model are found to be in a justifiable range, and no unnatural kinks have been observed.

[Figure]

**Figure R1:** Time variation of 10-m wind speed predicted from different similarity functions in the surface layer scheme of WRF model. The maximum value of wind speed in observational data is shown by dotted grey line.

[Figure]

**Figure R2:** Time variation of 2-m temperature predicted from different similarity functions in the surface layer scheme of WRF model.

*Comment 2: The paper is silent on the impact of potential clipping that is present in the WRF model. In many schemes the stability (psi) is kept in a certain range, as is the friction velocity, and some other parameters. Hence it is interesting to learn whether the WRF model got the complete freedom to show its sensitivity to the tested similarity functions. Hence please add some discussion to what is the range of -zeta the model could reach.*

**Reply:** Various surface layer schemes in different numerical models have restrictions on the values of the stability parameter ($\zeta$)/bulk Richardson number ($Ri_B$), as well as on the friction velocity and some other parameters. The present study utilizes the revised MM5 surface layer scheme (Jimenez et al., 2012), which is an updated version of the MM5 (fifth-generation Pennsylvania State University-National Centre for Atmospheric Research Mesoscale Model) surface layer scheme. It is observed that the MM5 scheme has several restrictions on the values of $\zeta$, $Ri_B$, friction velocity ($u_*$), and mean wind speed (U). In the MM5 surface layer scheme, U is restricted by a lower limit of 0.1 m s$^{-1}$ to control $Ri_B$ values from being inordinately high. The similarity functions for stable conditions are restricted by a limit of $-10$ on the values of both $\psi_m$ and $\psi_h$, and a limit on $\zeta$ ($> -10$) for unstable conditions is applied to prevent the use of similarity functions in strong stable and unstable conditions, respectively. Apart from this, a lower limit on $u_* (> 0.1 \text{ m s}^{-1})$ is also applied to control the value of heat flux from becoming zero in strong stable conditions.

On the other hand, in the revised MM5 surface layer scheme, most of these restrictions have been relaxed. For instance, the restrictions on both $\psi_m$ and $\psi_h$ in stable conditions as well as on $\zeta$ ($> -10$) in unstable conditions have been relaxed. This implies that the WRF model with the revised MM5 surface layer scheme has no restrictions on $\zeta$ or $Ri_B$ under stable as well as convective conditions and has complete freedom to show its sensitivity to the tested similarity functions. Moreover, the restriction on the values of $u_*$ is also reduced from 0.1 to 0.001 m s$^{-1}$ to allow smaller values of $u_*$, which can be common during the night. The restriction on the mean wind speed is as it is (i.e., U $> 0.1$ m s$^{-1}$) in the revised MM5 scheme.

As the reviewer has suggested, we have included a text regarding this in the revised version of the manuscript and presented it here for the reviewers' reference:

**Lines 107-110…**Note that the revised MM5 surface layer scheme has lower limits on the values of $u_* (> 0.001 \text{ m s}^{-1})$ and $U (> 0.1 \text{ m s}^{-1})$ that allow nocturnal values of $u_*$ at night and control $Ri_B$ values to be inordinately high, respectively (Jimenez et al., 2012). However, the stability parameter $\zeta$ or $Ri_B$ is not restricted in the revised MM5 surface layer scheme, which gives complete freedom to the WRF model to show its sensitivity to the tested similarity functions (Jimenez et al., 2012).

*Comment 3: There is some discussion about the free convection limit that could be added to the paper. On one hand the idea is that if the mean wind drops completely, then the CH should go to zero to allow the friction velocity to become zero too, so it disappears from the problem. However there are some other LES studies that show that despite the mean wind speed can drop to zero, the friction velocity will NOT drop to zero, i.e. that there is a*

*"minimum friction velocity" that is proportional the w\* (see Schumann 1980). Please discuss how the KY90 approach and implementation matches the minimum friction velocity approach.*

**Reply:** Studies reported in the literature suggest that friction velocity ($u_*$) cannot be zero when the mean wind drops to zero in free convective conditions; i.e., there should be a minimum friction velocity that is proportional to the $w_*$. We wish to highlight that the minimum value of $u_*$ is prescribed as $0.001$ m s$^{-1}$ in the existing version of the revised MM5 scheme based on the recommendations by Jimenez et al. (2012) to avoid the complexity that arises when mean wind drops to zero. Thus, the updated revised MM5 surface layer scheme with KY90 functional forms proposed in the present study also utilizes a minimum value of $u_*$ ($> 0.001$ m s$^{-1}$) as suggested by Jimenez et al. (2012).

As suggested by the reviewer, we have added text regarding this in the revised version of the manuscript and presented it here for the reviewers' reference:

**Lines 110-116…**Moreover, some of the LES studies reported in the literature suggest that the friction velocity cannot be zero when the mean wind drops to zero; i.e., there should be a minimum friction velocity that is proportional to the $w_*$ (Schumann, 1980). For this purpose, the existing version of the revised MM5 scheme sets $0.001$ m s$^{-1}$ as the minimum value of $u_*$ based on the recommendations by Jimenez et al. (2012). Thus, to avoid the complexity that arises when mean wind drops to zero, the updated revised MM5 scheme proposed in the present study also utilizes a minimum value of $u_*$ ($> 0.001$ m s$^{-1}$) as suggested by Jimenez et al. (2012) in the existing version of the revised MM5 scheme.

*Comment 4: I find the description of the observational site too limited. Please extend. What is the time frequency of the output of the obs? 10-min or 60 min? What is the vegetation of the measurement site? Idem for typical roughness length.*

**Reply:** The needful is done. The revised text is presented here for reviewers' reference:

**Lines 169-184…**For the evaluation of different simulations corresponding to newly installed similarity functions, observational data derived from the micrometeorological tower installed at Ranchi (India) has been utilized (Srivastava and Sharan, 2019; Srivastava et al., 2020; 2021). The dataset (Ranchi data) is derived from an instrument mounted on a 32-m tall tower at the Birla Institute of Technology Mesra in Ranchi, India with an average elevation of 609 m above sea level in a tropical region. The site has a few buildings in between east and northwest; agriculture land in between northwest and west; a residential area; and dense trees in between southeast and east. The site also has a relatively flat area in between southeast and west, which is free from any obstacle (Srivastava and Sharan, 2015). A fast-response sensor (CSAT3 Sonic Anemometer) at a height of 10 m with an average elevation of 609 m above sea level provides the temperature and the three components of wind at a 10 Hz frequency. The eddy covariance

technique (Stull 1988) is used to estimate heat and momentum fluxes at one-hour time resolution; however, the hourly temperature at 2 m is determined by averaging temperature observations available at a temporal scale of 1 minute from the slow response sensors located at logarithmic heights on the same tower. The time frequency of the output of the observation is 60 min. The roughness length for momentum ($z_0$) over the Ranchi domain is found to be around $0.009 \pm 0.007$ m during the summer, as suggested by Reddy and Rao (2016), who utilized the profile method to compute the values of $z_0$ based on the observed data from June 2011 to May 2012. However, we have also computed the value of $z_0$ based on the observational data utilized in the present study, but the value comes out to be higher than that suggested by Reddy and Rao (2016) and needs to be further validated.

***Comment 5: Concerning the real cases, it would be good to add some discussion about how many model grid cells are affected by the changed psi functions for how many time slots in the simulations, and in which weather regimes this occurs. That offers a more detailed insight in the modelling impacts.***

**Reply:** The present study is focused on evaluating the impacts of different similarity functions under convective conditions in the surface layer scheme of the WRF model. For this purpose, various functional forms of similarity functions have been newly installed in the surface layer scheme of the WRF model under convective conditions, and the similarity functions for stable stratification remain the same in all the experiments and CTRL simulation. This suggests that most of the changes due to different functional forms are expected to be visible in the convective regime (i.e., daytime). Due to this, we have considered the summer (MAM) season and analyzed the model output using different similarity functions in the WRF model for various variables during daytime only (i.e., unstable conditions). From Figure 4, it is observed that the differences between different similarity functions are more pronounced in strong unstable conditions. Thus, we have also analyzed the model output for various experiments during those hours in which strong convective conditions occur over most of the study domain.

Regarding the number of model grids affected by the changed similarity functions, we have shown the mean spatial distribution of model simulated variables and their differences with respect to the CTRL simulation utilizing the default version of the similarity functions in the revised MM5 scheme. For instance, a figure is also attached herewith, which shows the mean spatial differences of simulated ζ, $C_D$, and $C_H$ between CTRL simulation and other newly installed similarity functions. Note that no fixed pattern has been observed for the model grids that are being affected by the changed similarity functions; however, the changes are dependent on the considered variable and experiment. For instance, $\frac{z}{L}(= \zeta)$ simulated from the KY90 functional forms (Exp3) shows substantial differences over the whole study domain (i.e., all the model grids are affected) with respect to CTRL simulation (Figure R3).

[Figure]

**Figure R3:** Mean spatial distribution of model simulated $\zeta$ (1st row), $\mathbf{C_D}$ (3rd row) and $\mathbf{C_H}$ (5th row) from different experiments and their differences with respect to CTRL simulation averaged during daytime for whole simulation period. Hatched regions show significant differences at 95% confidence level in experiments with respect to CTRL simulation.

As per the reviewer's suggestion, text has been added to the revised version of the manuscript for better presentation and clarity of results. The modified text is presented here for reviewers' reference:

**Lines 409-414…**The results presented so far suggest that the changes corresponding to different functional forms of similarity functions under convective conditions in the surface layer parameterization of the WRF model are more pronounced in convective conditions during daytime hours. For the number of grid points over the study domain that are being affected by the changed similarity functions, no fixed pattern was observed; however, the changes depend on the considered variable and similarity functions. Furthermore, we observe that the changes are more pronounced in grids that experience strong instability during the daytime.

**Minor comments**

*Comment 1: $C_D$ and $C_H$ are never formally defined in the paper. I think it is good to add that for a more easy read.*

**Reply:** The mathematical expressions for both $C_D$ and $C_H$ are now added to the revised manuscript.

*Comment 2: Ln 28: tuned. I think this is not the right wording in the sense that to fit the relation between dimensionless groups, one must use observations*

**Reply:** The text has been modified accordingly in the revised version of the manuscript.

*Comment 3: Ln 84: Appendix B was referred to before Appendix A was referred to.*

**Reply:** The appendices A and B are interchanged in the revised text.

*Comment 4: Ln 86: …the CASES-99 dataset*

**Reply:** The necessary changes are made to the modified text.

*Comment 5: Equation 7: something seems to be missing between the brackets for the formula in the upper regime*

**Reply:** Equation 7 has been modified accordingly in the revised version of the manuscript.

*Comment 6: Equation 8: Same here, they look like loose hanging minuses.*

**Reply:** The needful is done.

*Comment 7: Ln 132: here the notations for phi's are suddenly in italic, while they are not in the rest of the manuscript so far.*

**Reply:** The phi's are accordingly changed throughout the text.

*Comment 8: Ln 169: For the computation, z is taken as 10 m and RiB is in the range $-2 \leq RiB \leq 0$. Can you justify the 10m and the RIB regime?*

**Reply:** We would like to point out that the turbulent measurements at both Ranchi (India) and the CASES-99 sites are used at a height of 10 m. Accordingly, $z$ is taken as 10 m for the calculation of $\zeta$, $C_D$, and $C_H$ in offline simulations. In principle, there are no restrictions on the $Ri_B$ values under convective conditions; however, for practical consideration, the range of $Ri_B$ is taken as $-2$ to $0$, which can cover all the different sublayers considered in convective conditions, from DNS ($\zeta > -0.04$) to FCS ($\zeta < -2$).

*Comment 9: Ln 347: typo in "moemntum"*

**Reply:** The needful is done.

*Comment 10: Ln 487: the bias is the mean of the difference between model and observations, so better to type the overbar over (p_i - o_i).*

**Reply:** It is corrected in the revised text.

*Comment 11: Figure 1: In the box for the stable boundary layer, "Change" should be "Cheng"*

**Reply:** "Change" has been replaced by "Cheng" in Figure 1.

*Comment 12: Figure 2: From these plots and the captions it is not directly clear which of the lines represents the new model implementation.*

**Reply:** The caption of Figure 2 has been modified accordingly and is presented here for reviewers' reference:

**Figure 2:** Integrated similarity functions $\psi_{m,h}(\zeta)$ for momentum and heat for default (F96; black line) and newly installed (BD71, CL73, and KY90; orange, gray, and blue lines) functions for unstable atmospheric surface layer.

*Comment 13: Figure 2: the caption says "default", but none of the labels in the figure indicates which of the four is the default.*

**Reply:** Figure 2 has been modified accordingly, and the caption of Figure 2 is presented in the previous reply.

***Comment 14: Figure 4: the legend box is overlying the vertically dashed lines three times***

**Reply:** It is corrected in the revised text.

***Comment 15: Figure 4: the caption is incomplete since the explanation is missing for DNS, DFS, FCS, DCS-FCS, DNS-DCS. I must say I find these graphs rather chaotic since these texts about the regimes are scattered all over the place. Can this not be solved by coloring the background of the diagram for the regimes in contrasting color. The caption is also incomplete since it does not explain what are Exp1-3. Better to label these BD71, CL73, and KY90. This applies to all figures afterwards.***

**Reply:** The caption of Figure 4 is modified in the revised version of the manuscript and presented here for reviewers' reference:

**Figure 4:** Variation of $\zeta$ with $Ri_B$ (upper panel), $C_D$ (middle panel), and $C_H$ (lower panel) with $\zeta$ calculated from bulk flux algorithm (offline simulation) for different functional forms of $\psi_m$ and $\psi_h$ corresponding to BD71, CL73, KY90, and F96 forms for smooth ($z_0 = 0.01$ m; 1st column), transition ($z_0 = 0.1$ m; 2nd column), and rough ($z_0 = 1$ m; 3rd column) surfaces. The background colour corresponds to different sublayers in convective conditions (Kader and Yaglom 1990), from the dynamic sublayer ($0 \geq \zeta > -0.04$; light gray) to the free convective sublayer ($\zeta < -2$; dark gray).

***Comment 16: Figure 4: headers: z0 must have a unit.***

**Reply:** The unit for $z_0$ is meter, and Figure 4 has been accordingly modified.

***Comment 17: Figure 4: the vertical axes have somewhat unnatural steps. Why not start at y=0?***

**Reply:** Since the differences in the simulated $\zeta$, $C_D$, and $C_H$ from different functional forms of similarity functions are smaller. So, for better visibility and clarity, we haven't started the y axis from zero for some of the subplots in Figure 4.

***Comment 18: Figure 8: RMSE must have a unit.***

**Reply:** Units have been included for RMSEs in Figure 8.

***Comment 19: All tables: Put the table caption above the caption.***

**Reply:** In all the tables, the captions are now moved from bottom to top in the revised version of the manuscript.

***Comment 20: Table 2: The number of decimals is really too large in this table. The typical measurement error of a temperature measurement including its representativeness error is about 0.3K, then 3 decimals for RMSE and MAE is really high. Friction velocity does not have more than 2 decimals significance, so 3 is too many here. Please reconsider also for the other variables, and in Table 3.***

**Reply:** Table 2 and 3 have been modified accordingly, and the values of bias, RMSE, and MAE for considered variables up to two decimal places are now used in the revised version of the manuscript.

**References:**

1. Dyer, A. J.: A Review of Flux-Profile Relationships, Boundary-Layer Meteorol., 7, 363–372. https://doi.org/10.1007/BF00240838, 1974.

2. Hong, S. Y., Noh, Y., & Dudhia, J.: A new vertical diffusion package with an explicit treatment of entrainment processes, Monthly weather review, 134, 2318-2341. https://doi.org/10.1175/1520-0493(2004)132<0103:ARATIM>2.0.CO;2, 2006.

3. Jiménez, P. A., Dudhia, J., González-Rouco, J. F., Navarro, J., Montávez, J. P., & García-Bustamante, E.: A Revised Scheme for the WRF Surface Layer Formulation, Mon. Wea. Rev., *140*, 898-918. https://doi.org/10.1175/MWR-D-11-00056.1, 2012.

4. Kader, B. A., and Yaglom, A. M.: Mean Fields and Fluctuation Moments in Unstably Stratified Turbulent Boundary Layers, Journal of Fluid Mechanics, 212, 637–662, https://doi.org/10.1017/S0022112090002129, 1990.

5. Reddy, N.N., Rao, K.: Roughness Lengths at Four Stations Within the Micrometeorological Network over the Indian Monsoon Region. Boundary-Layer Meteorology, **158**, 151–164. https://doi.org/10.1007/s10546-015-0080-2, 2016.

6. Schumann, U.: Minimum Friction Velocity and Heat Transfer in the Rough Surface Layer of a Convective Boundary Layer', Boundary-Layer Meteorol. 44, 311–326. 1988

7. Srivastava, P., and Sharan, M.: Characteristics of the Drag Coefficient over a Tropical Environment in Convective Conditions, J. Atmos. Sci., 72, 4903–4913, https://doi.org/10.1175/JAS-D-14-0383.1, 2015.

8. Srivastava, P., and Sharan, M.: Analysis of Dual Nature of Heat Flux Predicted by Monin-Obukhov Similarity Theory: An Impact of Empirical Forms of Stability Correction Functions, J. Geophys. Res. Atmos., 124, 3627–3646, https://doi.org/10.1029/2018JD029740, 2019.

9. Srivastava, P., Sharan, M., Kumar, M., and Dhuria, A. K.: On Stability Correction Functions over the Indian Region under Stable Conditions, Meteorol. Appl., 27:e1880, https://doi.org/10.1002/met.1880, 2020.

10. Srivastava, P., and Sharan, M.: Uncertainty in the Parameterization of Surface Fluxes under Unstable Conditions, J. Atmos. Sci., 78, 2237–2247, https://doi.org/10.1175/JAS-D-20-0350.1, 2021.

11. Stull, R. B.: An Introduction to Boundary Layer Meteorology, Kluwer Academic Publishers, Dordrecht, The Netherlands, 13, 670 pp, https://doi.org/10.1007/978-94-009-3027-8, 1988.

---

## Author Response (AR2)

**Response to the comments from Anonymous Referee#1**

We thank the reviewer for his/her critical evaluation of the manuscript, as well as his/her valuable comments and suggestions. A point-wise reply to the comments from the reviewer is given below:

**General Comment**

*The paper has been clearly improved. I still has some suggestions, most of them are minor points but three major issues remain. I still think that further English editing could improve the text. Examples are given as minor revisions below.*

**Reply:** We thank the reviewer for carefully going through the revised version of the manuscript and giving useful comments and suggestions, as well as encouraging remarks.

**Major Revisions**

*Comment 1: I am in doubt that z0 and zh are really the same in WRF. In this case the neutral Cd and Ch would be equal. But Figures 5(b) and (c) show that this is not the case.*

**Reply:** We agree with the reviewer that the values of $z_0$ and $z_h$ are not the same in the revised MM5 surface layer scheme available in the WRF model. In fact, in the model, the revised MM5 surface layer scheme uses a constant value of $z_0$, whereas the value of $z_h$ is deduced from the expression suggested by Brutsaert (1982).
We sincerely apologise for the confusion created in the previous reply.

The text regarding this has been modified in the revised version of the manuscript and is presented here for reviewer's reference:

**Lines 204-205…**Moreover, the scheme uses constant values of $z_0$, while the values of $z_h$ are calculated from the expression suggested by Brutsaert (1982).

*Comment 2: It is important that the values for z0 and zh used at the Ranchi station are given here, because only then the comparison between model and observations at Ranchi can be interpreted. I guess the value differs from the Ranchi value? This would explain the discrepancies between Cd in Figure 5 from the model and Figure 4 from the offline simulation. The error caused by different values of z0 (equivalent to neutral Cd) can be so large that the stability dependence by using different psi functions is less important. Nevertheless, the comparison with the Ranchi data is not useless because one gets an impression about the structural behavior of model results as a function of stratification compared with measurements. But these points must occur in the Discussion and Conclusion.*

**Reply:** We agree with the reviewer's concern. However, at the moment, due to the inaccessibility of long-term data on detailed surface properties such as vegetation structure needed to quantify the roughness length, we do not have an access to the precise values of $z_0$ and $z_h$ at the Ranchi station. Moreover, we wish to highlight that the values of $z_0$ and $z_h$ do not directly involve in the estimation of $C_D$, $C_H$, and the surface fluxes from the observational data, while they are important in computing these variables using the MOST framework. The error caused by different values of $z_0$ can be so large that the stability dependence of using different forms of similarity functions is less important in the computation of $C_D$ and $C_H$. As a result, three different values of $z_0$ have been chosen, similar to a recent study by Srivastava and Sharan (2021), which are representative of smooth ($z_0 = 0.01$ m), transition ($z_0 = 0.1$ m), and rough ($z_0 = 1.0$ m) surfaces to account for the impacts of using different $z_0$ on the estimation of $C_D$ and $C_H$ using various functional forms of similarity functions in offline simulations. In addition, the default value of $z_0$ is used in the revised MM5 surface layer scheme available in the WRF model, which is found to be approximately in the range $0.1 - 0.2$ m at the Ranchi station. Thus, one can interpret the results of $C_D$ and $C_H$ shown in Figures 4 and 5 from the offline simulation and the WRF model, respectively.

Figure 4 depicts the offline simulations with equal values of $z_0$ and $z_h$. While the revised version of the manuscript also discusses the results from the offline simulations with different values of $z_h$, assuming $z_0 = 0.1$ m. Figure R1 shows the variation of $\zeta$ with $Ri_B$, $C_D$, and $C_H$ with $\zeta$ calculated from the bulk flux algorithm using similarity functions corresponding to BD71, CL73, KY90, and F96 with different values of $z_h$ while $z_0$ is fixed. The values of $z_h$ are taken such that the ratio $\ln(z_0/z_h)$ assumes 0.1, 1, 2, 3, and 4. Figure R1 clearly shows that the estimated values of $\zeta$ are similar in near-neutral to moderately unstable conditions for all values of $z_h$; however, relatively smaller values have been found as the ratio $\ln(z_0/z_h)$ increases for each form of similarity function. Since the computation of $C_D$ does not involve the values of $z_h$ (Eqn. B9), the estimated values of $C_D$ for each form of similarity function are found to be approximately the same for different values of $z_h$. However, in the case of $C_H$, differences are clearly visible if one uses different values of $z_h$. The estimated $C_H$ using various similarity functions behaves similarly for different values of $z_h$, while the magnitude decreases as the ratio $\ln(z_0/z_h)$ increases.

The text has been added to discuss these issues in the revised version of the manuscript and is presented here for reviewer's reference:

**Lines 249-253…**Note that, the error caused by different values of $z_0$ can be so large that the stability dependence of using different forms of similarity functions is less important in the computation of $C_D$ and $C_H$. As a result, three different values of $z_0$ have been chosen, similar to a recent study by Srivastava and Sharan (2021), which are representative of smooth ($z_0 = 0.01$ m), transition ($z_0 = 0.1$ m), and rough ($z_0 = 1.0$ m) surfaces to account for the impacts of using different $z_0$ on the estimation of $C_D$ and $C_H$ from different functional forms of similarity functions in offline simulations.

**Lines 254-265…**Moreover, Figure 4 depicts the offline simulations with equal values of $z_0$ and $z_h$. While in the revised MM5 surface layer scheme available in the WRF model, the values of $z_0$ and $z_h$ are not the same. Thus, we have also attempted to discuss the results from the offline simulations with different values of $z_h$, assuming $z_0 = 0.1$ m. Figure S2 (supplementary material) shows the variation of $\zeta$ with $Ri_B$, $C_D$, and $C_H$ with $\zeta$ calculated from the bulk flux algorithm using similarity functions corresponding to BD71, CL73, KY90, and F96 with different values of $z_h$ while $z_0$ is fixed. The values of $z_h$ are taken such that the ratio $\ln(z_0/z_h)$ assumes 0.1, 1, 2, 3, and 4. Figure S2 clearly shows that the estimated values of $\zeta$ are similar in near-neutral to moderately unstable conditions for all values of $z_h$; however, relatively smaller values have been found as the ratio $\ln(z_0/z_h)$ increases for each form of similarity function. Since the computation of $C_D$ does not involve the values of $z_h$ (Eqn. B9), the estimated values of $C_D$ for each form of similarity function are found to be approximately the same for different values of $z_h$. However, in the case of $C_H$, differences are clearly visible if one uses different values of $z_h$. The estimated $C_H$ using various similarity functions behaves similarly for different values of $z_h$, while the magnitude decreases as the ratio $\ln(z_0/z_h)$ increases.

**Lines 283-295…**Note that, at the moment, due to the inaccessibility of long-term data on detailed surface properties such as vegetation structure needed to quantify the roughness length, we do not have an access to the precise values of $z_0$ and $z_h$ at the Ranchi station. Moreover, the values of $z_0$ and $z_h$ do not directly involve in the estimation of $C_D$, $C_H$, and the surface fluxes from the observational data, while they are important in computing these variables using the MOST framework. Thus, the default value of $z_0$ is used in the revised MM5 surface layer scheme available in the WRF model, which is found to be approximately in the range $0.1 - 0.2$ m at the Ranchi station. We wish to highlight that the $z_0$ used in the WRF model simulations at the Ranchi station is nearly similar to the case of $z_0 = 0.1$ m presented in Figure 4, and the offline simulations also indicate that the behaviour of the estimated $C_D$ and $C_H$ with $\zeta$ remains almost the same for different values of $z_0$ with slightly varying magnitudes. Thus, one can interpret the results of $C_D$ and $C_H$ shown in Figures 4 and 5 from the offline simulations and the WRF model, respectively, and can compare the WRF model simulated $C_D$ with the observed one at the Ranchi station. Although the model simulations and observed data may have a different $z_0$, the comparison of model simulated variables with the Ranchi data allows for an impression of the structural behaviour of model results as a function of stratification compared with measurements.

[Figure]

**Figure R1:** Variation of ζ with $Ri_B$ (upper panel), $C_D$ (middle panel) and $C_H$ (lower panel) with ζ calculated from bulk flux algorithm (offline simulation) for different functional forms of similarity functions corresponding to BD71, CL73, KY90, and F96 forms for different values of $z_h$ for the case when $z_0 = 0.1$ m. The background colour corresponds to different sublayers in convective conditions (Kader and Yaglom 1990), from the dynamic sublayer ($0 \geq ζ > -0.04$; light grey) to the free convective sublayer ($ζ < -2$; dark grey).

***Comment 3: It is important to know if the reference height for Cd and Ch in Figure 6 is also 10 m, or is the reference perhaps the height of the lowest model grid level? This needs to be said, otherwise the results cannot be interpreted.***

**Reply:** The transfer coefficients $C_D$ and $C_H$ shown in Figure 5 (not Figure 6) are at the reference height corresponding to the lowest model grid level, which is ~12 m in the present study. However, we have also attempted to analyze the variation of $C_D$ and $C_H$ at 10 m. Figure R2 shows the variation of $C_D$ and $C_H$ at 10 m simulated from the WRFv4.2.2 model using different forms of similarity functions corresponding to CTRL and Exp1-3 simulations. It is clear from Figure R2 that the variation of estimated $C_D$ and $C_H$ is almost similar to those presented in Figure 5. We present Figure R2 here solely for the reviewer's reference; it is not included in the manuscript.

A text has been added regarding this in the revised version of the manuscript and is presented here for reviewer's reference:

**Lines 331-333…**Note that the transfer coefficients $C_D$ and $C_H$ shown in Figure 5 are at the reference height corresponding to the lowest model grid level, which is ~12 m in the present study. However, we have also analyzed the behaviour of $C_D$ and $C_H$ at 10 m height with $\zeta$ and found that they behave similarly to those presented in Figure 5.

[Figure]

**Figure R2:** Variation of model simulated (a) $C_D$ and (b) $C_H$ at 10 m height with $\zeta$ from different experiments using different similarity functions corresponding to F96 (CTRL), BD71 (Exp1), CL73 (Exp2), and KY90 (Exp3) under convective conditions. The red circles in (a) denote the observed $C_D$ with $\zeta$ at the location of flux tower. The mean values of observed $C_D$ in each sublayer are shown with green solid circles along with standard deviations in the form of error bars. Depending upon the data availability, two or three bins of equal width are chosen in each sublayer. The background colour corresponds to different sublayers in convective conditions (Kader and Yaglom 1990), from the dynamic sublayer ($0 \geq \zeta > -0.04$; light grey) to the free convective sublayer ($\zeta < -2$; dark grey).

**Minor Revisions**

*Comment 1: Section 2.1 line 83: to improve the logic I suggest after the sentence ending with Sharan 2021). ....Following MOST they are formulated as ... and then come the equations (1) and (2). Below the sentence explaining the constants, you can add that further details of the Cd and Ch determination are given in Appendix A.*

**Reply:** This has been modified in the revised version of the manuscript and presented here for reviewer's reference:

**Lines 81-89…**The Monin-Obukhov similarity theory serves as the foundation for the surface layer parameterization (revised MM5 scheme) in the WRF model, and the surface turbulent fluxes are calculated based on the bulk approach using bulk transfer coefficients for momentum ($C_D$) and heat ($C_H$) (Namdev et al., 2024; Srivastava et al., 2021; Srivastava and Sharan, 2021). Following MOST, they are formulated as follows:

$$C_D = k^2 \left[ \ln\left(\frac{z + z_0}{z_0}\right) - \left\{ \psi_m\left(\frac{z + z_0}{L}\right) - \psi_m\left(\frac{z_0}{L}\right) \right\} \right]^{-2} \tag{1}$$

$$C_H = k^2 \left[ \ln\left(\frac{z + z_0}{z_0}\right) - \left\{ \psi_m\left(\frac{z + z_0}{L}\right) - \psi_m\left(\frac{z_0}{L}\right) \right\} \right]^{-1} \left[ \ln\left(\frac{z + z_h}{z_h}\right) \right.$$
$$\left. - \left\{ \psi_h\left(\frac{z + z_h}{L}\right) - \psi_h\left(\frac{z_h}{L}\right) \right\} \right]^{-1} \tag{2}$$

in which k is the von Karmann constant; $z_0$ and $z_h$ are the roughness lengths for momentum and heat, respectively; $\psi_m$ and $\psi_h$ are the integrated similarity functions for momentum and heat, respectively; and L is the Obukhov length scale.

Their determination based on MOST using integrated forms of the similarity functions is explained in Appendix B.

*Comment 2: Line 88: please correct to: ... in which k is the v. Karman constant.*

**Reply:** The needful is done by changing "a" to "the".

*Comment 3: Line 109/110: correct to: In this section, we briefly describe the implementation of different similarity functions for unstable stratification in the surface layer parameterization of WRFv4.2.2.*

**Reply:** The needful is done. The modified text is presented here for reviewer's reference:

**Lines 108-109…**In this section, we briefly describe the implementation of different similarity functions for unstable stratification in the surface layer parameterization of WRFv4.2.2.

*Comment 4: Line 110: I suggest writing: Note that two sets of functional forms, namely those suggested by Carl (1973) and the three…*

**Reply:** As suggested by the reviewer the sentence has been modified accordingly as:

**Lines 109-111…**Note that two sets of functional forms, namely those suggested by Carl et al. (1973) and the three sub-layer model proposed by Kader and Yaglom (1990) for convective conditions have not been included and tested in the surface layer scheme of the WRF modeling framework.

*Comment 5: Line 114: add reference for the KANSAS data.*

**Reply:** The following reference for the KANSAS data has been added to the reference list.

Izumi, Y.: Kansas 1968 Field Program Data Report. Bedford, MA, Air Force Cambridge Research Papers, No. 379, 79 pp, 1971.

*Comment 6: Line 133: still puzzling: it is still unclear which functions have been newly installed in the WRF model. In section 2.2.1 it is written that Businger (1971) was already used, but here and in Section 2.3 (lines 152, 153) it is stated that BD71, CL73 and KY90 are new. Does BD71 differ from Businger (1971)? This needs clarification.*

**Reply:** In the revised MM5 surface layer scheme, the similarity functions suggested by Businger et al. (1971), Carl et al. (1973), and the three sublayer model by Kader and Yaglom (1990) are newly installed. However, the functions proposed by Fairall et al. (1996) already exist.
      In addition, in Section 2.2.1, it is written that the BD71 functions already exist in the older version of the revised MM5 surface layer scheme (i.e., the MM5 scheme; Grell et al., 1994) in the WRF modeling system. However, the BD71 functions are not available in the revised MM5 scheme (Jimenez et al., 2012) considered in this study. In the updated version of the revised MM5 scheme described in this manuscript, the BD71 functions are added.

*Comment 7: Line 157: you mean here probably: ....in comparison to the other three functions (BD71,CL73,KY90) whose results are very similar to each other. ?*

**Reply:** The sentence has been modified accordingly as:

**Lines 154-156…**However, the rate of increase is slightly higher for F96 in comparison to the other three functions (BD71, CL73, and KY90), whose results are very similar to each other (Fig. 2b).

*Comment 8: Line 179: correct to: in the surface layer*

**Reply:** The needful is done.

*Comment 9: Line 192: it should be: 2x2 km2 and 6x6 km2,*

**Reply:** The needful is done.

*Comment 10: Lines 235/236: one can write simply: .... while they differ strongly from values obtained by Exp 3. Throughout the paper: the expression 'it has been observed' is often used, but this formulation is misleading, because 'observation' should be used in connection with measurements. To improve the text, one could write instead: 'it is found' or better, "one can see that" or "Figure .... shows that ......" or "one can see from Figure ... that ....."*

**Reply:** The corresponding text has been modified accordingly in the revised version of the manuscript as:

**Lines 228-230…**Moreover, results from the BD71, CL73, and F96 functions are even similar at higher instabilities (i.e., the whole range of $\zeta$ values), while they differ strongly from values obtained using the KY90 functions (Figure 4a-c).

Moreover, the expression 'it has been observed' has been improved throughout the text as per the reviewer's suggestion.

*Comment 11: Line 238: better write: consistent for all ratios z/z0...*

**Reply:** The corresponding sentence has been modified as:

**Lines 232-233…**This behaviour is found to be consistent for all ratios $z/z_0$ (Figures 4a-c) representative of smooth, transition, and rough surfaces.

*Comment 12: Line 242: Please check figure names in this paragraph, e.g. Figures 4b1, 4b2 do not exist.*

**Reply:** This is clarified in the text. Now the subplots in Figure 4 are referred to as 4a-i in the diagram as well as in the text.

*Comment 13: Lines 259/260: This is a repetition. If you want to summarize here, one can write To summarize, .....*

**Reply:** The corresponding text has been removed from the revised version of the manuscript.

*Comment 14: line 262: It is a pity. One could have added a figure showing this comparison using the observed roughness length.*

**Reply:** It is not feasible at the moment due to the inaccessibility of long-term data on detailed surface properties such as vegetation structure needed to quantify the roughness length. We do not have an access to the precise values of $z_0$ and $z_h$ at the Ranchi station.
The corresponding sentence has been removed in the revised version of the manuscript.

*Comment 15: Line 266: the flux tower*

**Reply:** The needful is done by replacing "a" by "the".

*Comment 16: Line 275: To make it clearer (see also my major revision) I suggest writing: Although the absolute values of the parameters differ from each other due to the different prescribed roughnesses the variation with Rib .... Is very similar as in the offline results .....*

**Reply:** The needful is done by modifying the corresponding text accordingly as:

**Lines 281-282…**Although the absolute values of the parameters differ from each other due to the different prescribed roughnesses, the variation of $\zeta$ with $Ri_B$, $C_D$ and $C_H$ with $\zeta$ is very similar to the offline results.

*Comment 17: Line 310-312: Considering Figure 5, it seems that this statement is not correct because in the neutral case (Rib going to zero) neutral values of Cd and Ch differ strongly from each other. I also do not think now that the different surface temperatures can explain the scatter in the modelled Ch. Please consider equations B8, B9, B10. For given zeta, z, z0 and zh there is only one value of Ch. There is no additional dependence on the surface temperature. Another reason must exist for the scatter and as I wrote in my first review, I still think that it might be due to the determination of zh in WRF, or there is a numerical reason (?)*

**Reply:** We thank the reviewer for this omission in the revised MM5 surface layer scheme available in the WRF modeling system, which uses different values of $z_0$ and $z_h$. Moreover, we agree with the reviewer that the relatively large scatter in the values of $C_H$ simulated by the WRF model can be attributed to the determination of $z_h$.

The text related to this has been modified in the revised version of the manuscript and is presented here for reviewer's reference:

**Lines 329-330…**The relatively large scatter in the values of $C_H$ simulated from the WRF model can be due to the parameterization of the ratio of momentum and scalar roughness lengths in the model.

*Comment 18: Line 360: correct to over the whole*

**Reply:** The needful is done.

**References:**

1. Brutsaert, W.: Evaporation into the Atmosphere: Theory, History, and Applications. Springer, Dordrecht, 299. http://dx.doi.org/10.1007/978-94-017-1497-6, 1982.
2. Businger, J. A., Wyngaard, J. C., Izumi, Y., & Bradley, E. F. 1971. "Flux-Profile Relationships in the Atmospheric Surface Layer in the Atmospheric Surface Layer". J. Atmos. Sci., 28(2), 181-189, https://doi.org/10.1175/1520-0469(1971)028<0181:FPRITA>2.0.CO;2, 1971.
3. Carl, D. M., Tarbell, T. C., and Panofsky, H. A.: Profiles of Wind and Temperature from Towers over Homogeneous Terrain, J. Atmos. Sci., 30, 788-794, http://dx.doi.org/10.1175/1520-0469(1973)030<0788:POWATF>2.0.CO;2, 1973.
4. Fairall, C. W., Bradley, E. F., Rogers, D. P., Edson, J. B., and Young, G. S.: Bulk Parameterization of Air-Sea Fluxes for Tropical Ocean global Atmosphere Coupled-Ocean Atmosphere Response Experiment, J. Geophys. Res., 101, 3747–3764, doi:10.1029/95JC03205, 1996.
5. Grell, G. A., Dudhia, J., & Stauffer, D.: A description of the fifth-generation Penn State/NCAR Mesoscale Model (MM5) (No. NCAR/TN-398+STR), University Corporation for Atmospheric Research. http://doi:10.5065/D60Z716B, 1994.
6. Izumi, Y.: Kansas 1968 Field Program Data Report. Bedford, MA, Air Force Cambridge Research Papers, No. 379, 79 pp, 1971.
7. Jiménez, P. A., Dudhia, J., González-Rouco, J. F., Navarro, J., Montávez, J. P., & García-Bustamante, E.: A Revised Scheme for the WRF Surface Layer Formulation, Mon. Wea. Rev., 140, 898-918, https://doi.org/10.1175/MWR-D-11-00056.1, 2012.
8. Kader, B. A., and Yaglom, A. M.: Mean Fields and Fluctuation Moments in Unstably Stratified Turbulent Boundary Layers, Journal of Fluid Mechanics, 212, 637–662, https://doi.org/10.1017/S0022112090002129, 1990.
9. Namdev, P., Srivastava, P., Sharan, M., & Mishra, S. K.: An Update to WRF Surface Layer Parameterization over an Indian Region, Dynam. Atmos. Ocean, 105, 101414, https://doi.org/10.1016/j.dynatmoce.2023.101414, 2024.
10. Srivastava, P., and Sharan, M.: Characteristics of the Drag Coefficient over a Tropical Environment in Convective Conditions, J. Atmos. Sci., 72, 4903–4913, https://doi.org/10.1175/JAS-D-14-0383.1, 2015.
11. Srivastava, P., Sharan, M., and Kumar, M.: A Note on Surface Layer Parameterizations in the Weather Research and Forecast Model, Dynam. Atmos. Ocean, 96, 101259, https://doi.org/10.1016/j.dynatmoce.2021.101259, 2021.
12. Srivastava, P., and Sharan, M.: Uncertainty in the Parameterization of Surface Fluxes under Unstable Conditions, J. Atmos. Sci., 78, 2237–2247, https://doi.org/10.1175/JAS-D-20-0350.1, 2021.